



# Bias Correction of Climate Models using a Bayesian Hierarchical Model

Jeremy Carter[1,2], Erick A. Chacón-Montalván[4], and Amber Leeson[2,3]

[1]Department of Mathematics and Statistics, Lancaster University, Lancaster, UK
[2]Data Science Institute, University of Lancaster, Lancaster, UK
[3]Lancaster Environment Center, University of Lancaster, Lancaster, UK
[4]Escuela Profesional de Ingeniería Estadística, Universidad Nacional de Ingeniería, Lima, Peru

**Correspondence:** Jeremy Carter (j.carter10@lancaster.ac.uk)

**Abstract.** Climate models, derived from process understanding, are essential tools in the study of climate change and its wide-ranging impacts on the biosphere. Hindcast and future simulations provide comprehensive spatiotemporal estimates of climatology that are frequently employed within the environmental sciences community, although the output can be afflicted with bias that impedes direct interpretation. Bias correction approaches using observational data aim to address this chal-

lenge. However, approaches are typically criticised for not being physically justified and not considering uncertainty in the correction. These aspects are particularly important in cases where observations are sparse, such as for weather station data over Antarctica. This paper attempts to address both of these issues through the development of a novel Bayesian hierarchical model for bias prediction. The model propagates uncertainty robustly and uses latent Gaussian process distributions to capture underlying spatial covariance patterns, partially preserving the covariance structure from the climate model which is based on

well-established physical laws. The Bayesian framework can handle complex modelling structures and provides an approach that is flexible and adaptable to specific areas of application, even increasing the scope of the work to data assimilation tasks more generally. Results in this paper are presented for one-dimensional simulated examples for clarity, although the method implementation has been developed to also work on multidimensional data as found in most real applications. Performance under different simulated scenarios is examined, with the method providing most value added over alternative approaches in the

case of sparse observations and smooth underlying bias. A major benefit of the model is the robust propagation of uncertainty, which is of key importance to a range of stakeholders, from climate scientists engaged in impact studies, decision makers trying to understand the likelihood of particular scenarios and individuals involved in climate change adaption strategies where accurate risk assessment is required for optimal resource allocation.



## 1 Introduction

Climate models are invaluable in the study of climate change and its impacts (Bader et al., 2008; Flato et al., 2013). Formulated from physical laws and with parameterisation and process understanding derived from past observations; climate models provide comprehensive spatiotemporal estimates of our past, current and future climate under different emission scenarios. Global climate models (GCMs) simulate important climatological features and drivers such as storm tracks and the El Niño–Southern Oscillation (ENSO) (Greeves et al., 2007; Guilyardi et al., 2009). In addition, independently developed GCMs agree on the

future direction of travel for many important features such as global temperature rise under continued net-positive emission scenarios (Tebaldi et al., 2021). However, GCMs are computationally expensive to run and also exhibit significant systematic errors, particularly on regional scales (Cattiaux et al., 2013; Flato et al., 2013). Regional climate models (RCMs) aim to dynamically downscale GCMs and more accurately represent climatology for specific regions of interest and have parameterisation, tuning and additional physical schemes optimised to the region (Giorgi, 2019; Doblas-Reyes et al., 2021). Despite this,

significant systematic errors remain, particularly for regions with complex climatology and with sparse in situ observations available to inform process understanding, such as over Antarctica (Carter et al., 2022). These systematic errors inhibit the direct interpretation of climate model output, particularly important in impact assessments (Ehret et al., 2012; Liu et al., 2014; Sippel et al., 2016).

There are many fundamental causes of systematic errors in climate models, including: the absence or imperfect represen-

tation of physical processes; errors in initialisation; influence of boundary conditions and finite resolution (Giorgi, 2019). The inherent complexity and computationally expensive nature of climate models makes direct reduction of systematic errors through model development and tuning challenging (Hourdin et al., 2017). End users are typically interested in only a narrow aspect of the output (e.g. possibly only one or two variables), which the model is unlikely to be specifically tuned for. Post-processing, bias correction techniques allow improvements to the consistency, quality and value of climate model output,

specific to the end user's focus of interest, with manageable computational cost and without requirement of in-depth knowledge behind the climate model itself (Ehret et al., 2012). Different end users are focused on different types of systematic errors, whether that's errors in the mean climatology, the multi-year trends or in other features of the output such as the covariance structure. This paper follows a common approach to focus on systematic errors in the parameters that describe the probability density function (PDF) at each site. Further, detailed discussion of this is given in Sect. 2 as are approaches to bias correction

within this context.

One of the fundamental issues often attached to bias correction is the lack of justification based on known physical laws and process understanding (Ehret et al., 2012; Maraun, 2016). Transfer functions are derived that are applied to the climate data to improve some aspect of consistency with observations, such as the mean in for example the delta method (Das et al., 2022) or the overall PDF in the case of quantile mapping (Qian and Chang, 2021). The spatiotemporal field and associated covariance

structure from the climate model, which is consistent with accepted physical laws, is typically not considered and so not preserved. Resulting corrected fields may exhibit too smooth or sharply varying behaviour over the region and discontinuities near observations. In addition, many approaches of bias correction fail to adequately handle uncertainties or estimate them



at all. Reliable uncertainty estimation is valuable for inclusion in impact studies to inform resulting decision making. This is especially true for regimes with tipping points, such as ice shelf collapse over Antarctica, where uncertainties in the climatology

can cause a regime shift (DeConto and Pollard, 2016).

In this paper a fully Bayesian approach using a hierarchical structure and latent Gaussian processes (GP) is proposed for bias correction, discussed in detail in Sect. 3. Parameter uncertainties are propagated through the model and the underlying covariance structure is derived both from observations and the climate model output. The approach is developed with the focus of applying bias correction to regions with sparse in situ observations, such as over Antarctica, where capturing uncertainty is

of key importance and where including data from all sources during inference is particularly valuable. In the method, climate model output is assumed to be generated from two underlying and independent stochastic processes, one relating to the true underlying field of interest (that also generates the in situ observations) and one that generates the bias present in the climate data. The aim is to separate these two processes and to infer their covariance structures. Posterior predictive estimates of the true underlying field across the region can then be made, which in turn can be used for bias correction. The ability of the

model in doing this depends on factors such as the density of observations and the relative smoothness of the truth and bias components. Simulated data is used to test the performance under scenarios with differing data density and latent covariance length scales, with results and discussion presented in Sect. 4.

The model is developed in a flexible Bayesian framework, where adjustments can easily be incorporated while maintaining robust uncertainty propagation. For example, extra predictors, such as elevation and latitude, can be included either in the

mean function or covariance matrix of the latent GPs. Alternatively, the model could be expanded to incorporate a temporal component of the bias accounting for variability across different seasons. This flexibility is important and increases the scope of the work, allowing the model to be applied to a wide range of scenarios, including for example application to many different meteorological fields and also combining observation data from different instruments rather than necessarily with respect to climate model output. Additionally, the Bayesian framework allows incorporation of domain specific, expert knowledge of

different data sources and their uncertainties through the choice of prior distributions.

## 2 Bias in Climate Models

Bias in climate models is defined in a number of similar ways across different papers. In Maraun (2016) it is defined as the systematic difference between any statistic derived from the climate model and the real-world true value of that statistic. While in Haerter et al. (2011), bias is defined as the time-independent part of the error between the climate model simulated values

and the observed values. In general, across the community involved with climate change impact studies, bias is used to refer to any deviation of interest between the model output and that of the true value (Ehret et al., 2012). Deviations of interest are typically with respect to the statistical properties of the data, for example the mean and variance as well as spatial properties such as the covariance length scale. The methodology developed in this paper treats bias with respect to deviations in the PDFs of the climate model output and observations at each site. Assuming a parametric form for the PDF, this translates to evaluating

bias in the parameters of the site-level PDFs, as discussed in Sect. 2.1. In order to model bias in real-world phenomena while



considering the intrinsic spatial structure, the parameters are allowed to vary spatially using stochastic processes, see Sect. 2.2. After evaluating bias across the domain, the methodology in this paper can be combined with current approaches of correcting bias in climate models, such as quantile mapping, discussed further in Sect. 2.3.

## 2.1 Bias in Random Variables

Consider a specific in situ observation site (e.g. an automatic weather station) with measurements of some variable $\boldsymbol{y} = [y_1, y_2, \ldots, y_n]$, such as midday temperature, and also comprehensive predictions from a climate model at the same location $\boldsymbol{z} = [z_1, z_2, \ldots, z_k]$. In this scenario, bias can be defined in terms of discrepancy between the PDFs of the in situ observations and the climate model predictions. In particular, assuming a parametric density function for both random variables, bias is translated to the discrepancy between the parameters of the PDFs. For example, assuming the observation measurements are independent
and identically distributed (i.i.d.) with normal distribution $Y \sim \mathcal{N}(\mu_Y, \sigma_Y)$ and the equivalent for the climate model outcomes $Z \sim \mathcal{N}(\mu_Z, \sigma_Z)$, then bias can be quantified by some discrepancy function of the mean parameters $(\mu_Z, \mu_Y)$ and the standard deviations $(\sigma_Z, \sigma_Y)$. This function can be defined in different ways, such as the difference $b(\mu_Z, \mu_Y) = \mu_Z - \mu_Y$ or the ratio $b(\sigma_Z, \sigma_Y) = \sigma_Z / \sigma_Y$.

## 2.2 Bias with Spatially Varying Parameters

Real-world applications, such as impact studies, typically require bias to be evaluated over a spatial region rather than just at a point. Consider a collection of $n$ observational sites $[\boldsymbol{y}_{s_1}, \ldots, \boldsymbol{y}_{s_n}]$, where for each site $i$ there exists $m$ daily measurements of some property such as midday temperature $\boldsymbol{y}_{s_i} = [y_{s_i,1}, \ldots, y_{s_i,m}]$. In addition, consider gridded output from a climate model of the same property at different locations $s^*$. In this scenario, instead of defining bias in terms of the discrepancy in the PDFs at a single point, bias can be defined with respect to the two latent spatial processes underlying the observed data $\{Y(s)\}$ and
the climate model output $\{Z(s)\}$. This allows bias to be estimated across the domain.

   As an example, assume that observations and the climate model output come from the spatial stochastic processes $\{Y(s) \sim \mathcal{N}(\mu_Y(s), \sigma_Y(s))\}$ and $\{Z(s) \sim \mathcal{N}(\mu_Z(s), \sigma_Z(s))\}$ respectively, where the distribution of data at each location $s$ is normal with spatially varying parameters $[\mu(s), \sigma(s)]$. The spatial structures of the latent processes are inherited from the spatial structures in the parameters, which are themselves modelled throughout the domain as spatial stochastic processes $\{\mu_Y(s)\}$, $\{\sigma_Y(s)\}$,
$\{\mu_Z(s)\}$ and $\{\sigma_Z(s)\}$. In this paper, GPs are used to model the spatial structures, which explicitly capture relationships for the expectation and covariance between points across the domain, see Sect. 3.3. Bias is then defined by some discrepancy function of these spatially varying parameters, such as $b(\mu_Z(s), \mu_Y(s)) = \mu_Z(s) - \mu_Y(s)$.

## 2.3 Bias Correction

   Climate model simulations are useful for impact assessments due to the their comprehensive spatiotemporal coverage and
ability to predict climate change signals, although bias in the output precludes direct interpretation (Ehret et al., 2012). Bias correction involves using observational data to predict and then reduce bias in the climate model output. Techniques vary in



focus and complexity, from using observations to apply an adjustment to the mean in the case of the delta change method (Das et al., 2022), to adjusting the whole PDF of the climate model output in the case of quantile mapping (Qian and Chang, 2021), and to approaches that use Generalised Additive Models (GAMs) to approximate transfer functions between the climate data

and the observed values (Beyer et al., 2020). Various studies compare relative performance between methods (Teutschbein and Seibert, 2012; Räty et al., 2014; Beyer et al., 2020; Mendez et al., 2020). Typically all approaches fail to capture uncertainty and explicitly model the spatial dependency between points and of the processes under study, not considering correlation between nearby measurements.

The approach proposed in this paper combines the use of a Bayesian hierarchical model for predicting bias across the region,
while explicitly modelling underlying spatial structures and capturing uncertainty, with the established approach of quantile mapping for applying the final correction to the climate model output. The details of the Bayesian hierarchical model are given in Sect. 3. The data is treated as generated from latent stochastic processes, as in Sect. 2.2, and estimates are made for parameters of the site-level PDFs of the observations and climate model output across the domain. This allows quantile mapping at each grid point of the climate model output to then be applied. Specifically, for each value of the time series
from the climate model output at a given point ($z_{s_i,j}$), this involves finding the percentile of that value and then mapping the value onto the corresponding value of the equivalent percentile of the PDF estimated from observational data. The cumulative density function (CDF) returns the percentile of a given value and the inverse CDF returns the value corresponding to a given percentile, which results in the following correction function $\hat{z}_{s_i,j} = F^{-1}_{Y_{s_i}}(F_{Z_{s_i}}(z_{s_i,j}))$, where $F_{,s_i}$ represents the CDF at the site $s_i$. The CDF can be estimated as an integral over the parametric form assumed for the PDF. The Bayesian hierarchical
model presented provides a collection of realisations for the PDF parameters at each site from an underlying latent distribution. Applying quantile mapping with each set of realisations then results in a collection of bias corrected time series, with an expectation and uncertainty.

The approach presented builds upon that of Lima et al. (2021) where the spatial dependency between in situ observations is modelled when estimating the PDF parameters and then qauntile mapping applied. In Lima et al. (2021) the spatial structure
in the climate model output is not explicitly modelled though, whereas in this paper the shared spatial structure between the observations and the climate model output is accounted for when conditioning, discussed further in Sect. 3.3. This is particularly important for regions of sparse in situ observations and results in conserving some of the information available from the climate model on the true spatial variation of parameters. Incorporating and partially conserving the spatial covariance structure of the climate model is desirable as it is derived from well established physical laws and reflects the assumption that the climate
model itself provides skill in assessing the site level parameters across the domain (Ehret et al., 2012). Additionally, in this paper the hierarchical model is embedded in a fully Bayesian framework and uncertainty in the PDF parameter estimates is propagated through the quantile mapping procedure to the bias corrected climate time series, which is missing from Lima et al. (2021).





## 3 Bias Prediction Methodology

The goal of the methodology developed and presented in this paper is to evaluate the bias in the climate model output across the domain in a framework that captures uncertainty robustly and that preserves information available from both the in situ observations and climate model output on underlying spatial structures. The resulting predictive bias can be coupled with known bias correction methods, such as quantile mapping, with the benefits of uncertainty quantification and inherited spatial structure. The overarching approach is summarised in Sect. 3.1 with a specific example given in Sect. 3.2. The properties of

GPs are discussed in Sect. 3.3.

### 3.1   Model Overview

In a probabilistic framework, the in situ observations and climate model output are treated as realisations from latent spatiotemporal stochastic processes, denoted as $\{Y(s,t) : s \in \mathcal{S}, t \in \mathcal{T}\}$ and $\{Z(s,t) : s \in \mathcal{S}, t \in \mathcal{T}\}$, respectively. Stochastic processes are sequences of random variables indexed by a set, which in this case are the spatial and temporal coordinates in the do-

main $(\mathcal{S}, \mathcal{T})$. A random variable is attributed to each spatiotemporal coordinate $(Y(s,t), Z(s,t))$. The data observed is then considered a realisation of the joint distribution over a finite set of random variables across the domain.

For the purpose of bias prediction, the random variables are treated as independent and identically distributed across time, such that for a given location $s$, $Y(s,t) \mid \boldsymbol{\phi}_Y(s) \overset{i.i.d.}{\sim} \mathcal{F}_Y(\boldsymbol{\phi}_Y(s))$ and $Z(s,t) \mid \boldsymbol{\phi}_Z(s) \overset{i.i.d.}{\sim} \mathcal{F}_Z(\boldsymbol{\phi}_Z(s))$, where $\mathcal{F}_Y(\cdot)$ and $\mathcal{F}_Z(\cdot)$ represent some generic site-level distributions with spatially varying vector parameters $\boldsymbol{\phi}_Y(s)$ and $\boldsymbol{\phi}_Z(s)$. This follows from

evaluating the time-independent component of the climate model bias. Consider evaluating bias in the values of January midday near-surface temperature over a region. While the values of nearby days are clearly dependent on each other, since focus is on evaluating time-independent bias, the time component of the data is dropped and only the marginal distribution considered. The marginal distribution in this case gives the probability of a certain value of January midday temperature just based on location and could for example be approximated as normal with mean and variance parameters, as mentioned in Sect. 2.1. In

the case of other climatological fields such as rainfall a more appropriate distribution might be that of a Bernoulli-Gamma with its own collection of parameters, as used in Lima et al. (2021). Caution in this treatment should be applied in cases where, for example, the observational site only has a limited number of days of data and these are bunched around the same relatively short time period, since this period is unlikely to be representative of the time series as a whole.

The disparity between the spatially varying parameters $\boldsymbol{\phi}_Y(s)$ and $\boldsymbol{\phi}_Z(s)$ in the site-level marginal distributions serves as

a measure of bias. Specifically, as in Sect. 2.2, the bias for each parameter $\phi_i$ can be defined by some discrepancy function $\phi_{B,i}(s) = b_i(\phi_{Y,i}(s), \phi_{Z,i}(s))$. Alternatively, the parameters associated with the climate model output $\phi_{Z,i}(s)$ can be defined as a function of the unbiased parameters $\phi_{Y,i}(s)$ and a latent bias function $\phi_{B,i}(s)$. In this paper an additive relationship is used, such that $\phi_{Z,i}(s) = \phi_{Y,i}(s) + \phi_{B,i}(s)$. Additionally, the bias $\phi_{B,i}(s)$ is considered independent of the value of $\phi_{Y,i}(s)$. To estimate the parameters across the domain and quantify the bias, these spatially varying parameters are modelled as spatial

stochastic processes with hyper-parameters $\boldsymbol{\theta}$. It's important to note that since the collection of parameters may not necessarily all belong to the same parameter space, their representation can be standardized by applying a link function transformation



to some of the parameters $\tilde{\phi}_i = h_i(\phi_i)$ so that all parameters can be represented by the same family of stochastic processes. In the methodology presented in this paper the family of Gaussian processes is used to model spatial dependencies. The full hierarchical model is then the following, with dependencies illustrated through the plate diagram shown in Fig. 1.


$$Y(s,t) \mid \boldsymbol{\phi}_Y(s) \overset{i.i.d.}{\sim} \mathcal{F}_Y(\boldsymbol{\phi}_Y(s)) \tag{1}$$

$$Z(s,t) \mid \boldsymbol{\phi}_Z(s) \overset{i.i.d.}{\sim} \mathcal{F}_Z(\boldsymbol{\phi}_Z(s)) \tag{2}$$

$$
\begin{cases}
\phi_{Z,i}(s) = \phi_{Y,i}(s) + \phi_{B,i}(s) & \text{if correct support,} \\
\phi_{Y,i}(s) \perp\!\!\!\perp \phi_{B,i}(s) \\
\phi_{Y,i}(s) \sim \mathcal{GP}(\cdot, \cdot \mid \boldsymbol{\theta}_{\phi_{Y,i}}) \\
\phi_{B,i}(s) \sim \mathcal{GP}(\cdot, \cdot \mid \boldsymbol{\theta}_{\phi_{B,i}}) \\
\\
\tilde{\phi}_{Z,i}(s) = \tilde{\phi}_{Y,i}(s) + \tilde{\phi}_{B,i}(s) & \text{if link function required for correct support.} \\
\tilde{\phi}_{Y,i}(s) \perp\!\!\!\perp \tilde{\phi}_{B,i}(s) \\
\tilde{\phi}_{Y,i}(s) \sim \mathcal{GP}(\cdot, \cdot \mid \boldsymbol{\theta}_{\tilde{\phi}_{Y,i}}) \\
\tilde{\phi}_{B,i}(s) \sim \mathcal{GP}(\cdot, \cdot \mid \boldsymbol{\theta}_{\tilde{\phi}_{B,i}})
\end{cases} \tag{3}
$$

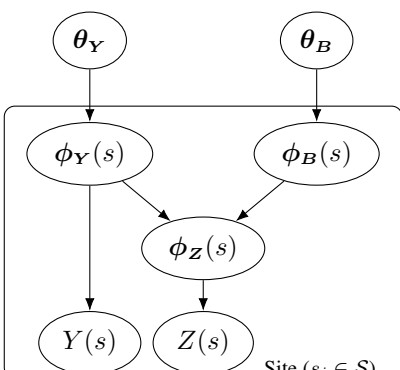

**Figure 1.** Plate diagram showing a generic version of the full hierarchical model. The in-situ observations $Y$ and climate model output $Z$ are generated from distributions with the collection of parameters $\boldsymbol{\phi}_Y$ and $\boldsymbol{\phi}_Z$ respectively. The parameters $\boldsymbol{\phi}_Z$ are modelled as some function of the parameters $\boldsymbol{\phi}_Y$ and some independent bias $\boldsymbol{\phi}_B$. The parameters $\boldsymbol{\phi}_Y$ and the corresponding bias $\boldsymbol{\phi}_B$ are each themselves modelled over the domain as generated from Gaussian processes with hyper-parameters $\boldsymbol{\theta}_Y$ and $\boldsymbol{\theta}_Z$.

Gaussian processes naturally introduce spatial structure into the parameters and enable inference with misaligned data.
Predictive estimates of the PDF parameters for each data source can be made for any set of locations across the domain.





Estimates at the climate model output locations are needed for bias correction, while there's also the possibility to compute
estimates at higher resolution and combine with a downscaling approach, as in Lima et al. (2021). Additional added benefits
of GPs include properties that facilitate inference, for example the additive property where the sum of two independent GPs is
itself also represented as a GP. More details following on from this and the application of GPs in the methodology is provided
in Sect. 3.3.

Inference on the parameters of site-level and spatial distributions of the model given the data is applied in a Bayesian hierar-
chical framework, where parameters of the model are treated as random variables with distributions. The distribution prior to
conditioning on any data is known as the prior distribution and allows the incorporation of a domain specific expert's knowl-
edge in the estimates of the parameters. The updated distribution after conditioning on the observed data is known as the
posterior and is approximated using Markov chain Monte Carlo (MCMC) methods, which provide samples from the posterior.
An important advantage of this framework is it allows flexible extensions of the model while automatically maintaining robust
uncertainty estimation. This results in the model being applicable to a wide range of problems and domains, especially impor-
tant for correcting climate model output since there's a broad range of users interested in different variables and domains with
varying levels of complexity.

## 3.2  Specific Model Example

Take the case of evaluating bias in the output of near-surface temperature from a climate model relative to some in situ
observations. The output from the in situ observations and the climate model are each considered as realisations from latent
spatiotemporal stochastic processes, $\{Y(s,t) : s \in \mathcal{S}, t \in \mathcal{T}\}$ and $\{Z(s,t) : s \in \mathcal{S}, t \in \mathcal{T}\}$ respectively. To evaluate bias the
time-independent marginal distributions are taken and the data treated as realisations from the spatial stochastic processes
$\{Y(s) : s \in \mathcal{S}\}$ and $\{Z(s) : s \in \mathcal{S}\}$. Temperature is known to have diurnal and seasonal dependency and so for the in situ
observation measurements to be representative of the time-independent marginal distribution there must be an equal spread of
the data over the time of day and season. To reduce this requirement the data can be filtered to just midday January values.
Filtering the data has the added benefit of simplifying the marginal distribution and so also the interpretation of bias, allowing
the bias to be evaluated for different seasons individually. In the case of January midday temperature, the site-level marginal
distributions can be approximated as normal, such that $Y(s) \sim \mathcal{N}(\mu_Y(s), \sigma_Y(s))$ and $Z(s) \sim \mathcal{N}(\mu_Y(s), \sigma_Y(s))$.

Treating the site-level distributions as normal results in bias being defined in terms of disparities in the mean and stan-
dard deviation parameters between in situ observations and climate model ouput, such that $\mu_B(s) = b_1(\mu_Y(s), \mu_Z(s))$ and
$\sigma_B(s) = b_2(\sigma_Y(s), \sigma_Z(s))$. Bias in the climate model output and the parameters of the in situ observations are considered
independent and both generated from separate spatial stochastic processes. For example, the bias in the mean $\mu_B(s)$ is con-
sidered independent of the mean of the in situ observations $\mu_Y(s)$ and both are modelled as generated from separate GPs:
$\mu_Y(s) \sim \mathcal{GP}(m_{\mu_Y}, k_{RBF}(s, s'|v_{\mu_Y}, l_{\mu_Y}))$ and $\mu_B(s) \sim \mathcal{GP}(m_{\mu_B}, k_{RBF}(s, s'|v_{\mu_B}, l_{\mu_B}))$. In this example the mean function of
the GP is considered a constant and the kernel/covariance function is considered a radial basis function parameterised by a
kernel variance and length scale. Defining the relationship $\mu_Z(s) = \mu_Y(s) + \mu_B(s)$ allows advantage of the property that the



sum of 2 independent GPs is itself a GP, such that $\mu_Z(s) \sim \mathcal{GP}(m_{\mu_Y} + m_{\mu_B}, k_{RBF}(s,s'|v_{\mu_Y}, l_{\mu_Y}) + k_{RBF}(s,s'|v_{\mu_B}, l_{\mu_B}))$. see

Sect. 3.3.

In the case of the standard deviation the parameter space ($\sigma(s) \in \mathbb{R}_{>0}$) is not the same as the sample space of a GP ($\mathbb{R}$) and so a link function is applied $log(\sigma(s)) = \tilde{\sigma}(s) \in \mathbb{R}$. The transformed parameters are then modelled as being generated from GPs: $\tilde{\sigma}_Y(s) \sim \mathcal{GP}(m_{\tilde{\sigma}_Y}, k_{RBF}(s,s'|v_{\tilde{\sigma}_Y}, l_{\tilde{\sigma}_Y}))$ and $\tilde{\sigma}_B(s) \sim \mathcal{GP}(m_{\tilde{\sigma}_B}, k_{RBF}(s,s'|v_{\tilde{\sigma}_B}, l_{\tilde{\sigma}_B}))$. To again take advantage of the property that the sum of 2 independent GPs is itself a GP, the relationship $\tilde{\sigma}_Z(s) = \tilde{\sigma}_Y(s) + \tilde{\sigma}_B(s)$ is defined. The parameter $\tilde{\sigma}_Z(s)$

is then distributed as: $\tilde{\sigma}_Z(s) \sim \mathcal{GP}(m_{\tilde{\sigma}_Y} + m_{\tilde{\sigma}_B}, k_{RBF}(s,s'|v_{\tilde{\sigma}_Y}, l_{\tilde{\sigma}_Y}) + k_{RBF}(s,s'|v_{\tilde{\sigma}_B}, l_{\tilde{\sigma}_B}))$. After predictions across the domain are made of the transformed parameter the inverse link function can be applied to get estimates of the non-transformed parameter.

The diagram in Fig.2 gives a representation of this full model in a hierarchical framework. Applying MCMC inference provides posterior realisations of the parameters of the model. This includes realisations from the posterior distribution of $\mu_Y$

and $\tilde{\sigma}_Y$ at all in situ observation locations and all climate model output locations, as well as realisations from the posterior of $\mu_B$ and $\tilde{\sigma}_B$ at all the climate model output locations. These realisations in addition to those of the parameters from the generating GPs can be used to compute the posterior predictive distribution of the parameters $[\mu_Y, \tilde{\sigma}_Y, \mu_B, \tilde{\sigma}_B]$ everywhere in the domain. For the purpose of applying bias correction, the posterior predictive distribution for these parameters can be evaluated at the locations of the climate model output. The parameters $\mu_Z$ and $\tilde{\sigma}_Z$ can then be computed and quantile mapping applied to

transform the predicted distribution of the climate model output onto that of the predicted distribution for in situ observations. Applying quantile mapping or alternative methods for multiple realisations of the parameters provides an expectation and uncertainty band for the bias corrected output.

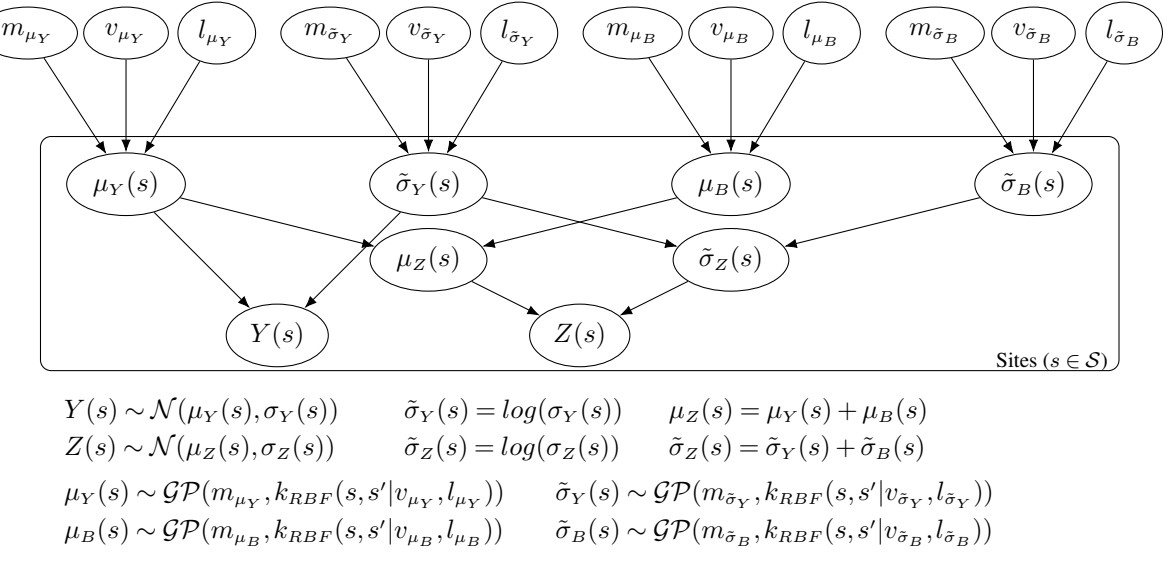

**Figure 2.** Plate diagram with directed acyclic graph showing the full hierarchical model for the case where the site-level distributions are assumed normal with parameters $\mu$ and $\sigma$. The distribution of these parameters across the domain is modelled with Gaussian processes.





### 3.3 Capturing Spatial Structure with Gaussian Processes

A collection of random variables $\boldsymbol{\phi} = [\phi_{s_1}, \phi_{s_2}, ..., \phi_{s_k}]$ indexed according to location in a domain can be modelled through a
spatial stochastic process, such as $\{\phi(s) : s \in \mathcal{S}\}$ (shorthand $\{\phi(s)\}$), where $\mathcal{S}$ represents the region under study. The family
of Gaussian processes (Rasmussen, 2004) have the property that any finite subset of random variables across the domain are
modelled as multivariate normal (MVN) distributed. Consider a collection of $k$ random variables, then the joint distribution
between these variables is MVN with $\boldsymbol{\phi} \sim \mathcal{N}_k(\boldsymbol{\mu}, \boldsymbol{\Sigma})$ where $\boldsymbol{\phi}$ is some $k$ dimensional random vector, $\boldsymbol{\mu}$ is some $k$ dimensional
mean vector and $\boldsymbol{\Sigma}$ is some $k \cdot k$ dimensional covariance matrix. Parameterising the mean and covariance of the MVN distri-
bution then gives the GP, which provides a distribution over continuous functions $\phi(s) \sim \mathcal{GP}(m(s), k(s, s'))$. The collection
of parameters for the mean and covariance functions are often referred to as hyper-parameters.

The mean function $m(s)$ of a GP gives the expectation of the parameter at the location index, allowing global relationships
for the variable given predictors. In this paper the mean function is considered as a constant across the domain for simplicity,
such that $m(s) = m$. In real-world applications a more complex relationship is likely to be useful, for example Eq. (4) assumes
a second order polynomial in two predictors, where the predictors $x_1(s)$ and $x_2(s)$ could be elevation and latitude.

$$m(s) = b_0 + b_1 \cdot x_1(s) + b_2 \cdot x_2(s) + b_3 \cdot x_1(s) \cdot x_2(s) + b_4 \cdot x_1(s)^2 + b_5 \cdot x_2(s)^2 = \boldsymbol{x}(s)^T \boldsymbol{\beta} \tag{4}$$

The kernel (covariance) function is typically some function of distance between points $d(\boldsymbol{s}, \boldsymbol{s}')$, parameterised by a length
scale $l$ and kernel variance $v$, for example Eq. (5) gives the well known Radial Basis Function (RBF) for the kernel. The
function of distance could be Euclidean or geodesic and arbitrarily complex, including factors such as wind paths, etc. The 2D
Euclidean case is given in Eq. (6), where predictors $x_3(s)$ and $x_4(s)$ could for example be latitude and longitude, which for
relatively small distances near the equator are approximately Euclidean. In Fig. 3, an example of how the covariance decays
with distance is given for the RBF kernel and realisations of a conditioned GP with the equivalent kernel are illustrated.

$$k_{RBF}(\boldsymbol{s}, \boldsymbol{s}') = v \cdot exp\left(-\frac{d(\boldsymbol{s}, \boldsymbol{s}')^2}{2l^2}\right) \tag{5}$$

$$d(\boldsymbol{s}, \boldsymbol{s}') = \sqrt{(x_3(\boldsymbol{s}') - x_3(\boldsymbol{s}))^2 + (x_4(\boldsymbol{s}') - x_4(\boldsymbol{s}))^2} \tag{6}$$

The kernel is often assumed stationary for simplicity, as in Lima et al. (2021), meaning that the relationship between covari-
ance and distance is consistent across the domain of study. This assumption is used in the methodology presented in this paper
and in the simulated examples given in Sect. 4. The validity of the stationarity assumption should be assessed on an application
basis, with factors such as complex topography contributing to non-stationarity.

Gaussian processes have the property that the sum of independent GPs is also a GP. This property is utilised in this paper as
the additive relationship $\phi_Z = \phi_Y + \phi_B$ is assumed, where $\phi_Y$ and the bias $\phi_B$ are taken as independent and generated from
latent Gaussian processes. Note that in the case of different supports between the parameter space and that of the sample space
of a Gaussian process, then a link function is included and the relationship $\tilde{\phi}_Z = \tilde{\phi}_Y + \tilde{\phi}_B$ assumed, where the parameters



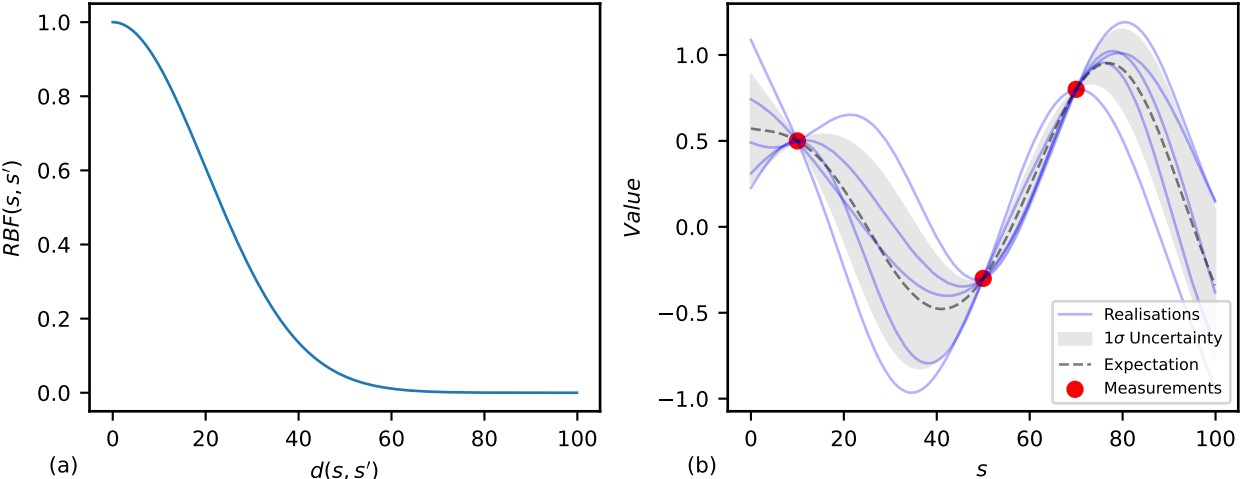

**Figure 3.** A) Values of the RBF function with a kernel variance equal to 1 and length scale equal to 20. B) Realisations of the GP with the equivalent kernel as in A and conditioned on 3 data points. The expectation and uncertainty of the distribution are shown.

$\tilde{\phi}_Y$ and $\tilde{\phi}_B$ are modelled as independent and generated from GPs. Assuming an additive relationship results in an easy to define distribution for $\phi_Z$ (or $\tilde{\phi}_Z$), which is a GP where the mean and covariances are simply the sum of the values from the independent GPs:

$$m_{\phi_Z} = m_{\phi_Y} + m_{\phi_B} \tag{7}$$

$$k_{\phi_Z}(s, s') = k_{\phi_Y}(s, s') + k_{\phi_B}(s, s') \tag{8}$$

This relationship captures the belief that the climate model output has some shared latent spatial covariance structure with the in situ observations but is inflicted by an independent bias. This relationship is shown graphically in Fig. 4. Additionally, an illustration for simulated realisations of $\phi_Y(s)$ and $\phi_B(s)$ from two underlying, independent latent GPs is provided in Fig. 5, where $\phi_Z(s)$ is also shown as the sum of the two realisations.

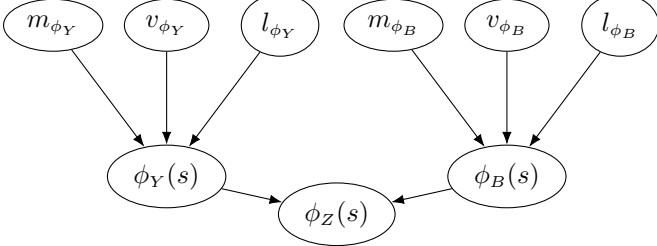

**Figure 4.** Directed acyclic graphs showing the joint dependency of the population parameters from the observations and the climate model.





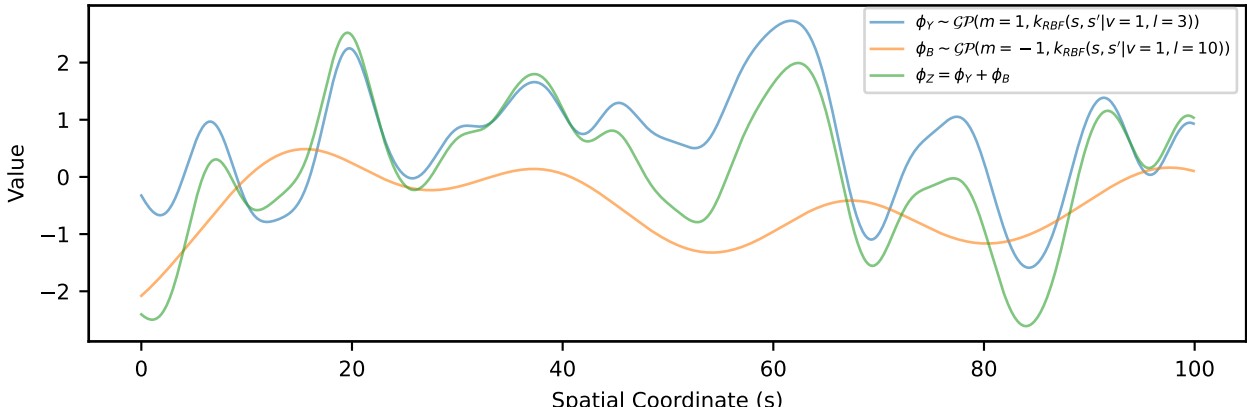

**Figure 5.** An illustration in 1 dimension of the parameters $\phi_Y$ and $\phi_B$ across some domain $0 \leq s \leq 100$. The parameters are realisations generated from GPs with different means and length scales. The parameter $\phi_Z$ is given as the sum of $\phi_Y$ and $\phi_B$.

In order to estimate the parameters relating to for example the in situ observations across the domain at some new locations $\phi_Y(\hat{s})$, conditioning is then performed on both the values of the parameter observed at the observation locations $\phi_Y(s^*)$ and the values of the parameter for the climate model output at the locations predicted by the climate model $\phi_Z(s')$.

## 4 Simulated Examples

The goal of the model is primarily to estimate, with reliable uncertainties, the true unbiased values of the PDF parameters at each location of the climate model output so bias correction can be applied. The model additionally infers the spatial structure of these parameters and their bias. Results are presented to highlight the advantage of two key features of the methodology over other approaches in the literature: modelling shared spatial covariance between the in situ data and climate model output through the inclusion of a shared generating latent process (Sect. 4.1) and the Bayesian hierarchical nature and uncertainty propagation (Sect. 4.2). One dimensional simulated examples are chosen for clarity in illustrating these features, although it is noted the implementation works for higher dimensional domains as is useful in real-world scenarios. The steps for generating the data and the results are presented separately for each example, while the discussion of results is done together in Sect. 5.

Inference of the parameters of the models is done in a Bayesian framework using MCMC and the No-U-Turn Sampler (NUTS) algorithm (Hoffman and Gelman, 2014) implemented in Numpyro (Phan et al., 2019). The parameters are treated as random variables with associated probability distributions. A prior distribution is set for each parameter and represents the belief on the distribution before observing any data, which typically incorporates knowledge from application specific experts. In the examples presented, relatively non-informative priors are chosen since the data is simulated and represents generic examples. The posterior distribution of each parameter is the updated distribution after observing and conditioning on the data. Estimates of the parameters $\phi_Y(s)$, $\phi_Z(s)$ and the corresponding bias $\phi_B(s)$ across the domain given the posterior and the observed data is then referred to as the posterior predictive.





## 4.1 Shared Latent Generating Processes: Non-Hierarchical Example

A non-hierarchical example is presented where direct measurements are assumed for one parameter of the PDFs for the in situ observations $\phi_Y(s)$ and for the climate model output $\phi_Z(s)$. The goal is to predict the parameter $\phi_Y(s)$ across the spatial
domain using information from both the simulated in situ observations and climate model output, which are related through $\phi_Z = \phi_Y + \phi_B$. The parameters $\phi_Y(s)$ and $\phi_B(s)$ are considered independent and generated from Gaussian processes. Comparison is made to the approach of inferring $\phi_Y(s)$ just from the in situ data, as in Lima et al. (2021). The purpose of this example is to illustrate the advantage of modelling shared latent generating processes between the observational data and the climate model output, as in Fig. 4. Relative performance is evaluated for three alternative simulated scenarios that correspond
to different possible real-world situations.

### 4.1.1 Simulated Data

The simulated data in this example is generated assuming the dependency in Fig. 4 and the relationship $\phi_Z = \phi_B + \phi_Y$, where $\phi_Y$ and $\phi_B$ are assumed independent. The latent Gaussian process distributions that generate $\phi_Y$ and $\phi_B$ across the domain are taken with constant mean and an RBF kernel (Eq. (5)). The hyper-parameters of these latent distributions and the number of
simulated observations are set for three scenarios, as given in Table 1. The prior distributions of the parameters are taken as the same for each scenario. Specifics of the data generation are given in Sect. B of the appendix.

    The three scenarios represent different potential real-world situations and the data generated for each is shown in Fig. 6. The first scenario (Fig. 6a) represents an example case where it is expected that there is ample data provided in the form of in situ observations to capture the features of the underlying complete realisation of $\phi_Y$ without significant added value provided
from inclusion of the climate model output during inference. The second scenario (Fig. 6b) is an adjustment where the in situ observations are relatively sparse and the underlying bias is relatively smooth. In this situation the climate model output should provide significant added value in estimating $\phi_Y$ across the domain since it is only afflicted by a comparatively simple bias that is easy to estimate. The final scenario (Fig. 6c) also involves sparse in situ observational data but with a reduced smoothness of the bias compared to other scenarios. In this scenario the climate model output should provide added value in estimating $\phi_Y$
across the domain but this will be limited compared to scenario two due to the difficulty of disaggregating and estimating the comparatively more complex bias.





|  | Scenario 1 | Scenario 2 | Scenario 3 |
|---|---|---|---|
| In-Situ Kernel Variance ($v_{\phi_Y}$) | 1.0 | 1.0 | 1.0 |
| In-Situ Kernel Lengthscale ($l_{\phi_Y}$) | 3.0 | 3.0 | 3.0 |
| In-Situ Mean Constant ($m_{\phi_Y}$) | 1.0 | 1.0 | 1.0 |
| In-Situ Observation Noise ($\sigma_{\phi_Y}$) | 0.1 | 0.1 | 0.1 |
| Bias Kernel Variance ($v_{\phi_B}$) | 1.0 | 1.0 | 1.0 |
| Bias Kernel Lengthscale ($l_{\phi_B}$) | 10.0 | 20.0 | 5.0 |
| Bias Mean Constant ($m_{\phi_B}$) | -1.0 | -1.0 | -1.0 |
| # In-Situ Observations | 80.0 | 20.0 | 20.0 |
| # Climate Model Predictions | 100.0 | 80.0 | 80.0 |

**Table 1.** A table showing the hyper-parameters of the two latent Gaussian processes used to generate the complete underlying realisations of $\phi_Y$, $\phi_B$ and hence $\phi_Z$, as well as observations of $\phi_Y$ and $\phi_Z$, on which inference is done for three scenarios. The number of observations representing in-situ data and climate model output is also given.

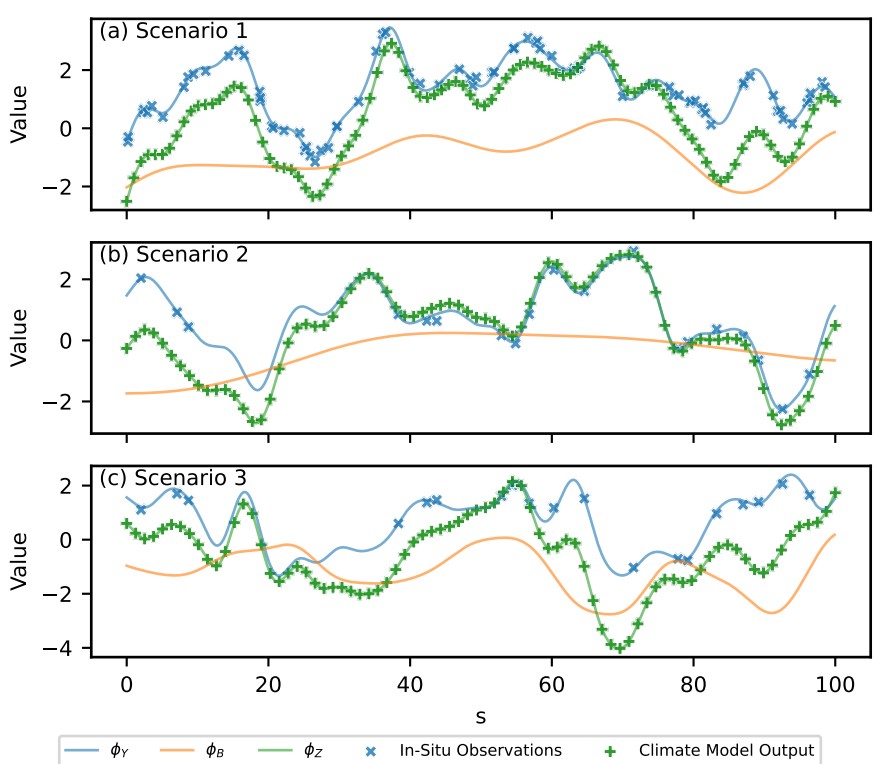

**Figure 6.** A figure showing simulated observed data for the parameters $\phi_Y$, $\phi_Z$ as well as the underlying latent functions for each parameter and the underlying bias $\phi_B$, defined through $\phi_Z = \phi_B + \phi_Y$. Three scenarios are shown and correspond to data generated from parameters in Table 1.



### 4.1.2 Results

The expectation, standard deviation and 95% credible intervals for the prior distribution and posterior distribution after inference of each parameter under the three different scenarios is given in Table 2. Comparisons are shown in the statistics between the posterior distributions of the full model presented in this paper, referred to as the shared process model, and the case where only the parameters for the in situ data are modelled as generated from a latent Gaussian processes, referred to as the single process model. In Sect. C of the appendix, an illustration is given of the prior and posterior distributions of each parameter after inference with the shared process model for scenario one.

Under all scenarios and for both the shared process and single process models the 95% credible interval of the posterior for every hyper-parameter bounds the value specified in generating the data. The expectation for the posterior distribution of the shared process model is in general closer to the specified value than the single process model and the range of the credible interval is smaller. In scenario one the differences between the models posteriors are relatively insignificant, although the shared process model does show a reduction in the uncertainty of the length scale for the latent process generating $\phi_Y$. In scenario two the difference is more significant and clear improvement is shown in both the expectation and uncertainty of latent parameter estimates for the shared process model. Improvement is also clear in estimates from the 3rd scenario, although the relative difference in performance between models is less significant.

Predictions for the underlying fields of the parameters $\phi_Y(s)$, $\phi_Z(s)$ and the corresponding bias $\phi_B(s)$ across the domain given the data, referred to as the posterior predictive, are shown in Fig. 7 for each scenario and for both the shared and single process models. The true underlying fields that the simulated observations were sampled from is also shown. The single process model is only concerned with estimating the underlying field of $\phi_Y(s)$ across the domain given observations of the parameter for the in situ data, so in Figs. 7a, 7c and 7e the climate model output and bias fields are excluded. To perform bias correction of the climate model output through quantile mapping, posterior predictive estimates of $\phi_Y(s)$ at the climate model output locations are required. The relative ability of the shared and single process models to estimate this is further assessed through $R^2$ scores, presented in Table 3.

In Fig. 7 it can be seen that the predictions of $\phi_Y(s)$ across the domain in the shared process case (Fig. 7b, 7d and 7f) are closer to the true underlying field and with smaller but still realistic uncertainty compared to the single process model. In scenario one, the difference between the posterior predictive distributions for $\phi_Y(s)$ across the domain between the two approaches is not substantial, with both models performing adequately, having $R^2$ scores of 0.99 and 0.97 respectively. In scenario two, the difference between estimates of $\phi_Y(s)$ between the models is significant with $R^2$ scores of 0.99 and 0.68 for the shared and single process models respectively. Finally, in scenario three there is again a significant difference in the estimates of $\phi_Y(s)$ between the models, with $R^2$ scores of 0.74 and 0.52 respectively, although the difference is reduced compared with scenario two.



| Scenario 1 | Specified | Prior Distribution | | | | Posterior Dist. (Shared Process) | | | | Posterior Dist. (Single Process) | | | |
|---|---|---|---|---|---|---|---|---|---|---|---|---|---|
| | Value | Exp. | Std. Dev. | C.I. L. | C.I. U. | Exp. | Std. Dev. | C.I. L. | C.I. U. | Exp. | Std. Dev. | C.I. L. | C.I. U. |
| In Situ Kernel Variance $v_{\phi_Y}$ | 1.0 | 0.67 | 0.67 | 0.02 | 2.46 | 1.25 | 0.30 | 0.73 | 1.86 | 1.04 | 0.31 | 0.57 | 1.69 |
| In Situ Kernel Lengthscale $l_{\phi_Y}$ | 3.0 | 15.00 | 8.66 | 3.09 | 36.12 | 2.96 | 0.06 | 2.85 | 3.08 | 2.73 | 0.20 | 2.32 | 3.10 |
| In Situ Mean Constant $m_{\phi_Y}$ | 1.0 | 0.00 | 2.00 | -3.92 | 3.92 | 1.14 | 0.28 | 0.61 | 1.68 | 1.23 | 0.26 | 0.74 | 1.76 |
| In Situ Observation Noise $\sigma_{\phi_Y}$ | 0.1 | 2.00 | 2.00 | 0.05 | 7.38 | 0.11 | 0.01 | 0.09 | 0.12 | N/A | N/A | N/A | N/A |
| Bias Kernel Variance $v_{\phi_B}$ | 1.0 | 15.00 | 8.66 | 3.09 | 36.12 | 2.10 | 1.30 | 0.48 | 4.72 | N/A | N/A | N/A | N/A |
| Bias Kernel Lengthscale $l_{\phi_B}$ | 10.0 | 0.00 | 2.00 | -3.92 | 3.92 | 11.45 | 1.28 | 9.07 | 14.00 | N/A | N/A | N/A | N/A |
| Bias Mean Constant $m_{\phi_B}$ | -1.0 | 0.25 | 0.14 | 0.01 | 0.49 | -1.00 | 0.64 | -2.31 | 0.24 | N/A | N/A | N/A | N/A |
| **Scenario 2** | Specified | Prior Distribution | | | | Posterior Dist. (Shared Process) | | | | Posterior Dist. (Single Process) | | | |
| | Value | Exp. | Std. Dev. | C.I. L. | C.I. U. | Exp. | Std. Dev. | C.I. L. | C.I. U. | Exp. | Std. Dev. | C.I. L. | C.I. U. |
| In Situ Kernel Variance $v_{\phi_Y}$ | 1.0 | 0.67 | 0.67 | 0.02 | 2.46 | 1.13 | 0.28 | 0.66 | 1.66 | 1.49 | 0.53 | 0.65 | 2.55 |
| In Situ Kernel Lengthscale $l_{\phi_Y}$ | 3.0 | 15.00 | 8.66 | 3.09 | 36.12 | 2.97 | 0.06 | 2.86 | 3.09 | 3.70 | 0.44 | 2.83 | 4.56 |
| In Situ Mean Constant $m_{\phi_Y}$ | 1.0 | 0.00 | 2.00 | -3.92 | 3.92 | 0.70 | 0.27 | 0.15 | 1.22 | 0.69 | 0.40 | -0.14 | 1.44 |
| In Situ Observation Noise $\sigma_{\phi_Y}$ | 0.1 | 2.00 | 2.00 | 0.05 | 7.38 | 0.12 | 0.03 | 0.08 | 0.18 | N/A | N/A | N/A | N/A |
| Bias Kernel Variance $v_{\phi_B}$ | 1.0 | 15.00 | 8.66 | 3.09 | 36.12 | 1.24 | 0.99 | 0.16 | 3.23 | N/A | N/A | N/A | N/A |
| Bias Kernel Lengthscale $l_{\phi_B}$ | 20.0 | 0.00 | 2.00 | -3.92 | 3.92 | 23.69 | 5.79 | 12.29 | 34.90 | N/A | N/A | N/A | N/A |
| Bias Mean Constant $m_{\phi_B}$ | -1.0 | 0.25 | 0.14 | 0.01 | 0.49 | -0.66 | 0.64 | -1.87 | 0.62 | N/A | N/A | N/A | N/A |
| **Scenario 3** | Specified | Prior Distribution | | | | Posterior Dist. (Shared Process) | | | | Posterior Dist. (Single Process) | | | |
| | Value | Exp. | Std. Dev. | C.I. L. | C.I. U. | Exp. | Std. Dev. | C.I. L. | C.I. U. | Exp. | Std. Dev. | C.I. L. | C.I. U. |
| In Situ Kernel Variance $v_{\phi_Y}$ | 1.0 | 0.67 | 0.67 | 0.02 | 2.46 | 1.18 | 0.33 | 0.62 | 1.83 | 0.85 | 0.33 | 0.30 | 1.50 |
| In Situ Kernel Lengthscale $l_{\phi_Y}$ | 3.0 | 15.00 | 8.66 | 3.09 | 36.12 | 3.00 | 0.07 | 2.87 | 3.14 | 3.08 | 0.49 | 2.03 | 3.96 |
| In Situ Mean Constant $m_{\phi_Y}$ | 1.0 | 0.00 | 2.00 | -3.92 | 3.92 | 0.95 | 0.30 | 0.35 | 1.53 | 0.90 | 0.29 | 0.33 | 1.48 |
| In Situ Observation Noise $\sigma_{\phi_Y}$ | 0.1 | 2.00 | 2.00 | 0.05 | 7.38 | 0.16 | 0.06 | 0.03 | 0.27 | N/A | N/A | N/A | N/A |
| Bias Kernel Variance $v_{\phi_B}$ | 1.0 | 15.00 | 8.66 | 3.09 | 36.12 | 1.50 | 1.02 | 0.28 | 3.56 | N/A | N/A | N/A | N/A |
| Bias Kernel Lengthscale $l_{\phi_B}$ | 5.0 | 0.00 | 2.00 | -3.92 | 3.92 | 6.34 | 1.71 | 3.23 | 9.20 | N/A | N/A | N/A | N/A |
| Bias Mean Constant $m_{\phi_B}$ | -1.0 | 0.25 | 0.14 | 0.01 | 0.49 | -1.17 | 0.50 | -2.11 | -0.10 | N/A | N/A | N/A | N/A |

**Table 2.** A table showing summary statistics for the prior and posterior distributions including the expectation (Exp.), standard deviation (Std. Dev.) and lower and upper bounds for the 95% credible interval (C.I. L. and C.I. U.). The posterior distributions for the shared and single process models are given. The specified value for each parameter used to generate the data is also shown.

| | Shared Process Model | | Single Process Model | |
|---|---|---|---|---|
| | Exp. | Std.Dev. | Exp. | Std.Dev. |
| Scenario 1 | 0.99 | 0.00 | 0.97 | 0.01 |
| Scenario 2 | 0.99 | 0.01 | 0.68 | 0.07 |
| Scenario 3 | 0.74 | 0.12 | 0.52 | 0.10 |

**Table 3.** A table showing the expectation and standard deviation of $R^2$ scores for the posterior predictive estimates of $\phi_Y$ at the climate model output locations for the shared and single process models for each scenario.





**Figure 7.** Expectation and $1\sigma$ uncertainty of the posterior predictive distributions across the domain for the parameter $\phi_Y(s)$ and the corresponding bias $\phi_B(s)$ for three scenarios. The underlying latent functions that the data are measurements of is included.





## 4.2 Bayesian Framework: Hierarchical Example

A hierarchical example is presented in this section where the in situ data and climate model output are simulated at each site
as generated from normal distributions, as in the specific example given in Sect. 3.2. The goal of the model is the same as in
Sect. 4.1, that is to predict the parameters of the PDFs for the climate model output and in situ observations at the locations
of the climate model output. An example of how uncertainty in these predictions can be propagated through bias correction
techniques such as quantile mapping is then presented. The purpose of this section is to demonstrate the model working in
the intended hierarchical structure and to illustrate the benefit of having a fully Bayesian hierarchical model for uncertainty
estimation.

### 4.2.1 Simulated Data

The simulated data in this example is generated assuming the dependencies in Sect. 3.2 and Fig. 2. Defining $Y(s,t)$ and $Z(s,t)$
as the in-situ data and climate model output respectively, then the time-independent PDF at each site is taken as normal such
that $Y(s) \sim \mathcal{N}(\mu_Y(s), \sigma_Y(s))$ and $Z(s) \sim \mathcal{N}(\mu_Y(s), \sigma_Y(s))$. The following relationship is assumed for the mean parameters
$\mu_Z(s) = \mu_Y(s) + \mu_B(s)$, where $\mu_B(s)$ is the bias in the mean for the climate data. For the standard deviation, the parameters
are first transformed using a logarithmic link function and then the relationship $\tilde{\sigma}_Z(s) = \tilde{\sigma}_Y(s) + \tilde{\sigma}_B(s)$ is assumed, where
$\tilde{\sigma}_B(s)$ is the bias in the transformed parameter. The latent distributions that generate $\mu_Y(s)$, $\mu_B(s)$, $\tilde{\sigma}_Y(s)$ and $\tilde{\sigma}_B(s)$ across
the domain are assumed as independent GPs with constant mean and an RBF kernel. The hyper-parameters for these latent
generating processes are set for a single scenario, as given in Table 4. Further specifics of the data generation is provided in
Sect. B of the appendix.

There are 40 locations corresponding to simulated in situ observation sites, where for each site 20 measurements are gener-
ated. Likewise, there are 80 locations corresponding to simulated climate model output and at each location 100 samples are
generated. This reflects the typical scenario where the climate model output has greater spatiotemporal coverage than in situ
observations but is also afflicted with greater bias. In Fig. 8 examples of the generated samples are shown corresponding to the
nearest sites for three locations. It is clear that, due to limited observations, there will be significant uncertainty in estimates
of the mean and standard deviation parameters at each site and it's important this uncertainty is propagated when estimating
the parameters across the domain. The underlying, complete realisations of the parameters $\mu_Y(s)$, $\mu_Z(s)$, $\sigma_Y(s)$ and $\sigma_Z(s)$, as
well as the bias $\mu_B(s)$ and $\sigma_B(s)$, are shown in Fig. 9. In addition, the mean value and standard deviation of the generated data
is given at the simulated in situ observation and climate model sites.




|                                                                        | Hierarchical Scenario |
| ---------------------------------------------------------------------- | --------------------- |
| In-Situ Mean, Kernel Variance ($v_{\mu_Y}$)                            | 1.0                   |
| In-Situ Mean, Kernel Lengthscale ($l_{\mu_Y}$)                        | 3.0                   |
| In-Situ Mean, Mean Constant ($m_{\mu_Y}$)                             | 1.0                   |
| In-Situ Transformed Variance, Kernel Variance ($v_{\tilde{\sigma}^2_Y}$) | 1.0                 |
| In-Situ Transformed Variance, Kernel Lengthscale ($l_{\tilde{\sigma}^2_Y}$) | 3.0             |
| In-Situ Transformed Variance, Mean Constant ($m_{\tilde{\sigma}^2_Y}$) | 1.0                  |
| Bias Mean, Kernel Variance ($v_{\mu_B}$)                              | 1.0                   |
| Bias Mean, Kernel Lengthscale ($l_{\mu_B}$)                           | 10.0                  |
| Bias Mean, Mean Constant ($m_{\mu_B}$)                                | -1.0                  |
| Bias Transformed Variance, Kernel Variance ($v_{\tilde{\sigma}^2_B}$) | 1.0                   |
| Bias Transformed Variance, Kernel Lengthscale ($l_{\tilde{\sigma}^2_B}$) | 10.0              |
| Bias Transformed Variance, Mean Constant ($m_{\tilde{\sigma}^2_B}$)   | -1.0                  |
| # Spatial Locations of In-Situ Observations                            | 40.0                  |
| # Spatial Locations of Climate Model Predictions                       | 80.0                  |
| # Samples per Location of In-Situ Observations                         | 20.0                  |
| # Samples per Location of Climate Model Predictions                    | 100.0                 |

**Table 4.** A table showing the hyper-parameters used to generate the complete underlying realisations and the measurement data on which inference is done for the hierarchical scenario. The number of sites where data is generated along with the number of samples for each site is also given.

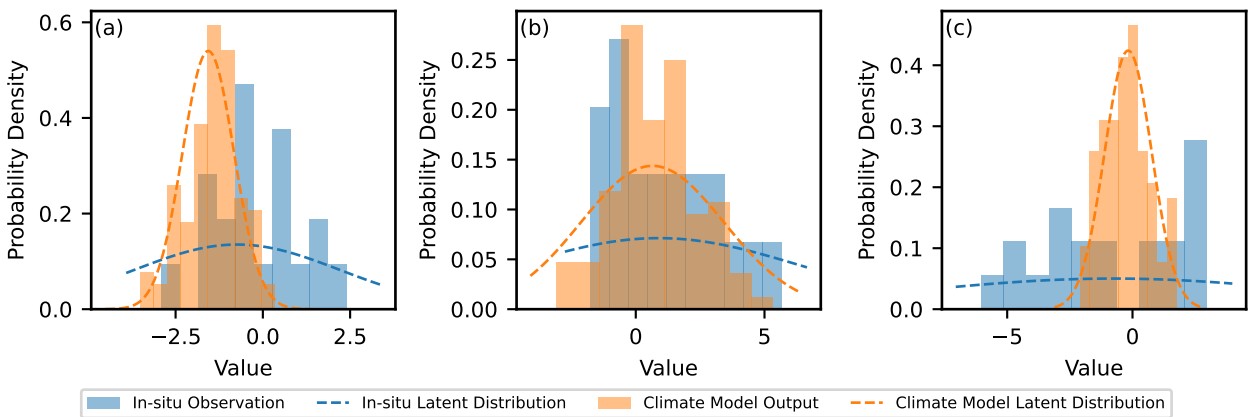

**Figure 8.** Histograms for the climate model output at three locations and the corresponding data from the nearest in situ observation site. The locations are a) s=11.4, b) s=46.8 and c) s=79.7. The latent normal distribution the data was generated from is illustrated as a dotted line.





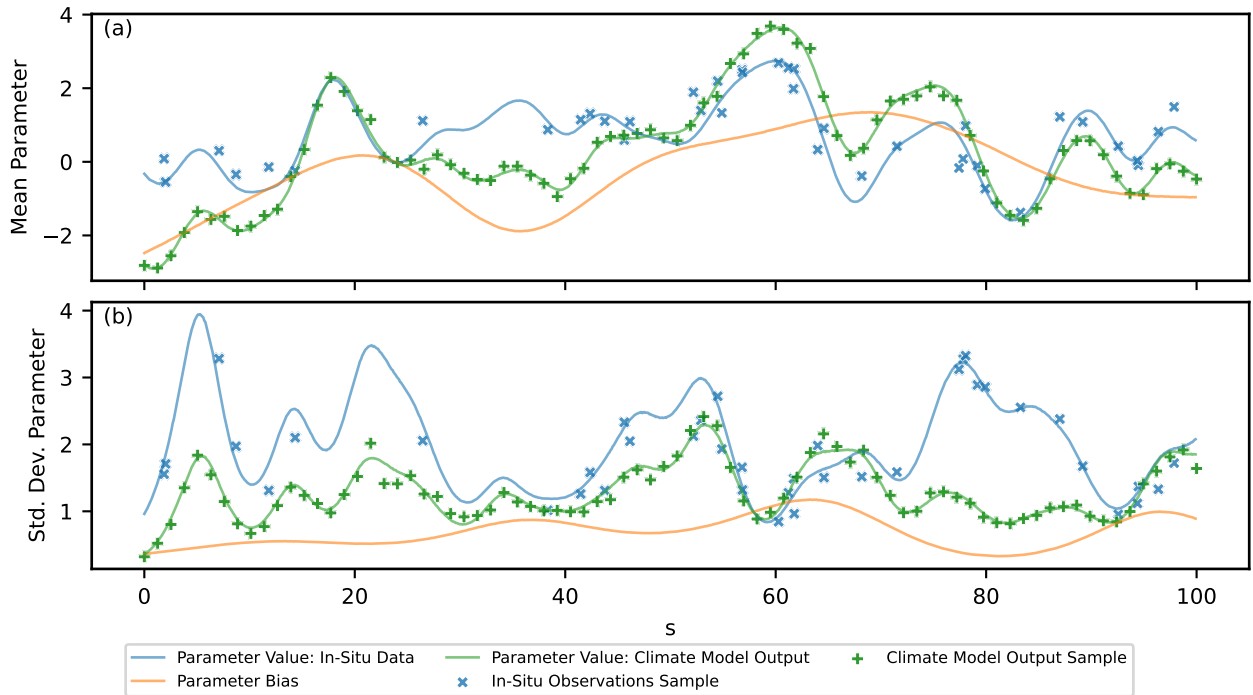

**Figure 9.** Simulated underlying functions for the parameters $\mu_Y(s)$, $\mu_B(s)$, $\mu_Z(s)$, $\tilde{\sigma}_Y(s)$, $\tilde{\sigma}_B(s)$ and $\tilde{\sigma}_Z(s)$ as well as the values at the observation locations for the in situ and climate model data.

### 4.2.2 Results

The expectation, standard deviation and 95% credible intervals for the prior and posterior distributions of each parameter are given in Table 5. The 95% credible interval of the posterior for every hyper-parameter bounds the value specified in generating the data. As expected the posterior distribution for each parameter is concentrated closer to the value specified when generating the data than the relatively non-informative prior distributions. The prior and posterior distributions for each parameter are plot in Fig. C2 of the appendix.





| Hierarchical Scenario | | Prior Distribution | | | | Posterior Distribution | | | |
|---|---|---|---|---|---|---|---|---|---|
| | Specified Value | Exp. | Std. Dev. | C.I. L. | C.I. U. | Exp. | Std. Dev. | C.I. L. | C.I. U. |
| In-Situ Mean, Kernel Variance $v_{\mu_Y}$ | 1.0 | 0.67 | 0.67 | 0.02 | 2.46 | 1.00 | 0.32 | 0.49 | 1.63 |
| In-Situ Mean, Kernel Lengthscale $l_{\mu_Y}$ | 3.0 | 15.00 | 8.66 | 3.09 | 36.12 | 3.00 | 0.22 | 2.56 | 3.43 |
| In-Situ Mean, Mean Constant $m_{\mu_Y}$ | 1.0 | 0.00 | 2.00 | -3.92 | 3.92 | 0.73 | 0.28 | 0.17 | 1.26 |
| In-Situ Transformed Variance, Kernel Variance $v_{\tilde{\sigma}_Y^2}$ | 1.0 | 0.67 | 0.67 | 0.02 | 2.46 | 0.70 | 0.25 | 0.30 | 1.17 |
| In-Situ Transformed Variance, Kernel Lengthscale $l_{\tilde{\sigma}_Y^2}$ | 3.0 | 15.00 | 8.66 | 3.09 | 36.12 | 2.94 | 0.24 | 2.47 | 3.40 |
| In-Situ Transformed Variance, Mean Constant $m_{\tilde{\sigma}_Y^2}$ | 1.0 | 0.00 | 2.00 | -3.92 | 3.92 | 1.12 | 0.24 | 0.66 | 1.61 |
| Bias Mean, Kernel Variance $v_{\mu_B}$ | 1.0 | 0.67 | 0.67 | 0.02 | 2.46 | 1.38 | 0.63 | 0.42 | 2.58 |
| Bias Mean, Kernel Lengthscale $l_{\mu_B}$ | 10.0 | 15.00 | 8.66 | 3.09 | 36.12 | 12.02 | 3.59 | 5.08 | 18.50 |
| Bias Mean, Mean Constant $m_{\mu_B}$ | -1.0 | 0.00 | 2.00 | -3.92 | 3.92 | -0.78 | 0.56 | -1.89 | 0.29 |
| Bias Transformed Variance, Kernel Variance $v_{\tilde{\sigma}_B^2}$ | 1.0 | 0.67 | 0.67 | 0.02 | 2.46 | 0.92 | 0.48 | 0.24 | 1.86 |
| Bias Transformed Variance, Kernel Lengthscale $l_{\tilde{\sigma}_B^2}$ | 10.0 | 15.00 | 8.66 | 3.09 | 36.12 | 8.97 | 1.96 | 5.07 | 12.58 |
| Bias Transformed Variance, Mean Constant $m_{\tilde{\sigma}_B^2}$ | -1.0 | 0.00 | 2.00 | -3.92 | 3.92 | -0.86 | 0.42 | -1.73 | -0.06 |

**Table 5.** A table showing summary statistics for the prior and posterior distributions including the expectation (Exp.), standard deviation (Std. Dev.) and lower and upper bounds for the 95% credible interval (C.I. L. and C.I. U.). The specified value for each parameter used to generate the data is also shown.

The posterior predictive estimate for the underlying fields of $\mu_Y(s)$, $\mu_B(s)$, $\sigma_Y(s)$ and $\sigma_B(s)$ across the domain given the data is shown in Fig. 10. The true underlying fields of the parameters are also shown, as are the mean and standard deviation values of the samples of simulated in situ observations and climate model outputs at the locations where they are sampled. The posterior predictive appears to perform well at capturing the spatial features of the underlying fields while also exhibiting a reasonable one sigma uncertainty range that bounds the majority of the underlying function. For example, in the range of $s \in [15, 25]$, where the main data source is the biased climate model output, the prediction accurately captures features of the true, unobserved latent mean $\mu_Y(s)$ and standard deviation $\sigma_Y(s)$. Uncertainty in the parameters of $\mu_Y(s)$ and $\sigma_Y(s)$ at the observation sites, due to limited samples, is propagated through the model. This is reflected in the uncertainty shown in estimates of the posterior predictive at the observation sites.

Bias correction of samples from the climate model output for a single site is shown in Fig. 11. The site chosen is at $s = 11.4$ and is the same as in Fig. 8a. A generic time series for the climate model output and in situ observations is generated from the correct mean and standard deviations of the samples. Quantile mapping of the climate model time series is performed for each posterior predictive realisation of $\mu_Y(s)$, $\mu_Z(s)$, $\sigma_Y(s)$ and $\sigma_Z(s)$. This results in multiple realisations of bias corrected time series with an expectation and uncertainty.



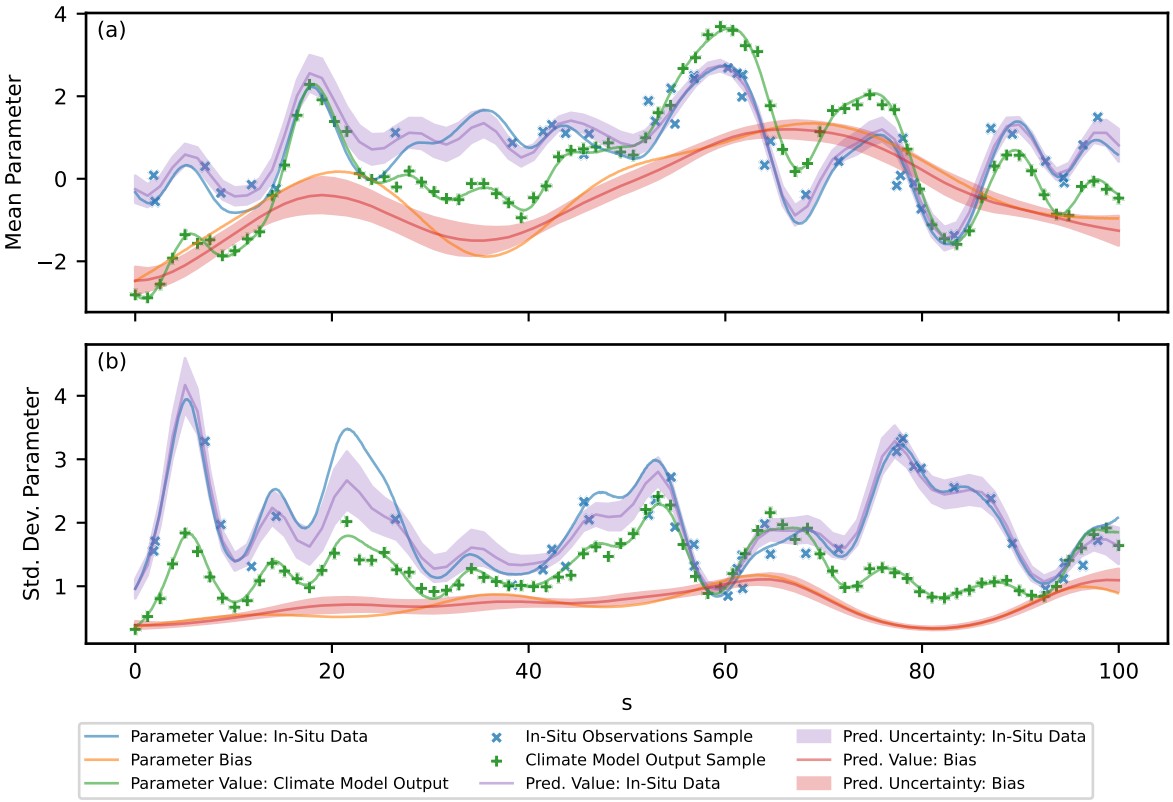

**Figure 10.** A figure showing the expectation and one sigma uncertainty of the posterior predictive distribution across the domain for the parameters $\mu_Y(s)$, $\mu_B(s)$, $\sigma_Y(s)$ and $\sigma_B(s)$ as well as their true underlying latent functions.

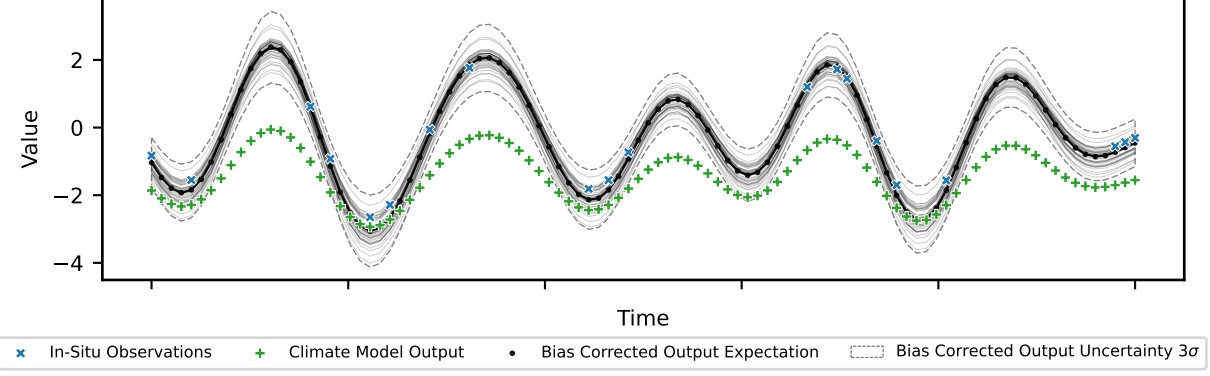

**Figure 11.** Simulated time series for the climate model output at location $s = 11.4$ and for the nearest in situ observation site. Realisations of the climate model bias corrected time series are shown along with the expectation and three sigma uncertainty range.



## 5 Discussion

The methodology presented in this paper assumes that each spatially varying parameter of the PDF for the climate model output is generated from two independent, latent GPs. One of these latent processes is also modelled to generate the equivalent parameter for the PDF of the in situ data, while the other latent process generates the bias. This reflects the belief that the climate model provides skillful estimates of these parameters across the domain and that the spatial covariance structure, generated from equations based on established physical laws, has similar features to the true underlying structure. The climate model output, while afflicted with bias, has comprehensive spatiotemporal coverage and provides useful information in the inference of the true values of the parameters across the domain, assuming the bias signal can be adequately deconstructed from the climate model output with the use of in situ observations. Incorporating this into the model presented in this paper provides added value over approaches where the parameters $\phi_Y(s)$ are inferred from the in situ observations alone, as in Lima et al. (2021). This is demonstrated in Sect. 4.1, where added value is assessed for three scenarios with differing density of observations and complexity of the bias signal.

Added value is assessed with respect to: summary statistics for the posterior distributions of the parameters of the latent GPs, visual examination of the expectation and standard deviation for posterior predictive estimates of the site-level PDF parameters across the domain, and comparison of $R^2$ scores for the PDF parameter $\phi_Y(s)$ at the locations of the climate model output. In Sect. 4.1.2 it is shown that most added value is provided, across all these measures, in the case of scenario two, where in situ observations are sparse compared to the climate model output and the underlying bias is relatively smooth compared to the unbiased signal. The bias can be estimated with high accuracy and precision, despite sparse in situ observations, since it varies smoothly across the domain, which also means the climate model output can be disaggregated and the unbiased component estimated across the domain with high accuracy and precision.

It is also shown in Sect. 4.1.2 that if the density of in situ observations is increased to similar levels as the climate model output itself, then the value added from the climate model output in inference of the unbiased parameters is reduced. This is demonstrated in scenario one, where the number of in situ observations is sufficient to adequately capture the spatial features of the underlying process (Fig. 7) as well as the latent spatial covariance structure, encoded through the hyper-parameter estimates of the latent GP (Table 2). Additionally, if the complexity of the bias signal is increased, through for example reducing the length scale of the latent generating process, then again added value is reduced. This is shown in scenario three, where a relatively more complex bias compared with scenario two makes it more difficult to disaggregate the climate model output into its biased and unbiased components, thus reducing the benefit provided in estimates of $\phi_Y(s)$. It is noted that, while added value is reduced relative to scenario two, benefits are still shown for scenarios one and three and incorporating the climate model output in inference improves overall performance.

In addition to shared latent processes, another important feature of the methodology presented in this paper is the Bayesian framework, where the parameters of the model are treated as random variables with associated distributions. This framework is flexible and allows for robust uncertainty propagation, which is important for making the model applicable to a wide range of real-world applications where bias prediction is required. Additionally, expert knowledge can be incorporated in the inference





through the choice of prior distributions, which is especially important where the data is sparse. In Sect. 4.2 results for a

simulated one dimensional hierarchical example illustrate uncertainty propagation between parameter values of the PDF at each sample site and the values of the hyper-parameters of the latent generating processes. Uncertainty present in the different levels of the hierarchical model are incorporated in the final posterior predictive estimates of the PDF parameters across the domain. These posterior predictive estimates can be used in bias correction techniques, such as quantile mapping, which is illustrated in Fig. 11. This results in multiple realisations of the final bias corrected time series, with an expectation and

uncertainty range. Robust uncertainty computation that incorporates the spatial relationships between points is important for impact assessments and resulting decision making. Having multiple realisations for the final bias corrected time series is also useful for further propagation of uncertainty in process models driven by climate model output, such as land surface models Liu et al. (2014).

## 6   Conclusion

Current approaches for bias prediction and correction do not aim to preserve the spatial covariance structure of the climate model output (Ehret et al., 2012). Climate models are fundamentally based on established physical laws and so the covariance structures are desirable since it is reasonable to assume that they are physically realistic. In addition, current approaches typically either neglect uncertainty or inadequately model uncertainty propagation through the model. In this paper a fully Bayesian hierarchical model for bias correction is presented where latent GP distributions are used to capture and preserve underlying

covariance structures. The Bayesian nature allows robust uncertainty propagation under a flexible modelling framework where the model is easily expanded for specific real-world scenarios, increasing the scope of the work.

Simple simulated examples are chosen to illustrate the key features of the model. In Sect. 4.1, results are displayed for a non-hierarchical example where the focus is on illustrating the nature of GPs and how assuming a shared latent GP between the in situ data and climate model output allows inference on the unbiased field from both sources of data. This is shown to

be particularly important in the case of sparse data and a simple bias, where the climate model output provides significant value added in predictions. In Sect. 4.2, results are presented for a hierarchical case and focus is on illustrating how the model propagates uncertainty between the different levels and to the final parameter predictions that are used in bias correction. Uncertainty in the parameter estimates is easily propagated in bias correction of the time series from the climate model at every location through the existing approach of quantile mapping. This results in a bias corrected time series with uncertainty

bands, which is desirable for use in impact studies that compute predictions on responses to climate change and for informing decisions based on these. This is especially true in areas where the climatology is hard to model and in situ observations are sparse, such as Antarctica, meaning the uncertainty is expected to be significant (Carter et al., 2022).

The model presented is a step towards adequately capturing uncertainty and incorporating underlying spatial covariance structures from the climate model in bias correction. The primary limitation is the assumption that the spatial structure of the

site-level parameters can be adequately modelled through a stationary GP. Over large and complex topographic regions it is likely that the covariance length scale will vary across the domain and this is something that will need assessing for each



specific application. Additionally, many real-world applications will necessitate specific model adjustments, such as incorporating a mean function dependent on factors like elevation and latitude, handling non-Gaussian data, and accounting for other bias structures. Since the model is developed in a Bayesian framework and inference on the parameters conducted with

MCMC, model adjustments are simple to incorporate with adequate uncertainty propagation. The next step then is to apply the methodology to a real-world dataset, incorporating additional modelling components and further exploring advantages as well as limitations that arise.

*Code and data availability.* The code used to generate the simulated data, fit the model, make predictions and create the figures/tables is available at: https://doi.org/10.5281/zenodo.10053653 (Carter, a).

The data used to create the plots is available at: https://doi.org/10.5281/zenodo.10053531 (Carter, b).





## Appendix A:  Posterior and Posterior Predictives

### A1  Full Hierarchical Model

The in situ observations and climate model output are treated as realisations from the stochastic processes $\{Y(s)\}$ and $\{Z(s)\}$ respectively, where the random variables for a given site are distributed as:

$$Y(s) \sim \mathcal{F}(\boldsymbol{\phi}_Y(s)) \tag{A1}$$

$$Z(s) \sim \mathcal{F}(\boldsymbol{\phi}_Z(s)) \tag{A2}$$

The symbols $\boldsymbol{\phi}_Y(s)$ and $\boldsymbol{\phi}_Z(s)$ represent the collection of parameters that describe the PDF at the site. For example if the PDF is normal, then $\boldsymbol{\phi}_Y(s) = [\mu_Y(s), \sigma_Y(s)]$ and $\boldsymbol{\phi}_Z(s) = [\mu_Z(s), \sigma_Z(s)]$.

Consider a collection of $n_Y$ in situ observational sites, where for each site $i$ there exists $m_i$ measurements of some property.
In addition, consider gridded output from a climate model at $n_z$ locations, where at each location there exists $m_z$ measurements of the same property. The data can then be represented through the following:

$$\boldsymbol{y} = [\boldsymbol{y}_{s_1}, \ldots, \boldsymbol{y}_{s_{n_y}}] \tag{A3}$$

$$\boldsymbol{y}_{s_i} = [y_{s_i,1}, \ldots, y_{s_i,m_i}] \tag{A4}$$

$$\boldsymbol{z} = [\boldsymbol{z}_{s_1}, \ldots, \boldsymbol{z}_{s_{n_z}}] \tag{A5}$$

$$\boldsymbol{z}_{s_j} = [z_{s_j,1}, \ldots, z_{s_j,m_z}] \tag{A6}$$

Also, defining the collection of in situ observation sites as $\boldsymbol{s}_y = [s_1, \ldots, s_{n_y}]$ and the collection of climate model output locations as $\boldsymbol{s}_z = [s'_1, \ldots, s'_{n_z}]$, then the collection of PDF parameter values for each set of locations is written as:

$$\boldsymbol{\phi}_Y(\boldsymbol{s}_y) = [\boldsymbol{\phi}_Y(\boldsymbol{s}_1), \ldots, \boldsymbol{\phi}_Y(\boldsymbol{s}_{n_y})] \tag{A7}$$

$$\boldsymbol{\phi}_Z(\boldsymbol{s}_z) = [\boldsymbol{\phi}_Z(\boldsymbol{s}'_1), \ldots, \boldsymbol{\phi}_Z(\boldsymbol{s}'_{n_z})] \tag{A8}$$

The PDF parameters are themselves each modelled as being generated from latent stochastic processes $\{\phi_Y(s)\}$ and $\{\phi_Z(s)\}$. The latent processes that generate the parameters for climate model are considered composed of two independent processes, one that also generates the equivalent parameters for the in situ observations and another that generates some bias, such that $\phi_Z(s) = \phi_Y(s) + \phi_B(s)$. The family of GPs are chosen for the latent processes. A link function is used for the case where the parameter space is not the same as the sample space for GPs. Considering the case of no link function, the following
can be be written:



$$\phi_Y(s) \sim \mathcal{GP}(\cdot, \cdot | \boldsymbol{\theta}_{\phi_Y}) \tag{A9}$$

$$\phi_B(s) \sim \mathcal{GP}(\cdot, \cdot | \boldsymbol{\theta}_{\phi_B}) \tag{A10}$$

$$\phi_Z(s) \sim \mathcal{GP}(\cdot, \cdot | \boldsymbol{\theta}_{\phi_Y}, \boldsymbol{\theta}_{\phi_B}) \tag{A11}$$

The collection of hyper-parameters for the generating processes are given by $\boldsymbol{\theta}_{\phi_Y}$ and $\boldsymbol{\theta}_{\phi_B}$ respectively. Note the additive property of GPs allows $\phi_Z(s)$ to also be represented by a GP, where the mean and covariances are computed from the sum of the relative values from the independent processes. The posterior distribution for the model can then be written as:

$$P(\boldsymbol{\phi}_Y(\boldsymbol{s}_y), \boldsymbol{\phi}_Z(\boldsymbol{s}_z), \boldsymbol{\theta}_{\phi_Y}, \boldsymbol{\theta}_{\phi_B} | \boldsymbol{y}, \boldsymbol{z}) = \frac{P(\boldsymbol{y}, \boldsymbol{z} | \boldsymbol{\phi}_Y(\boldsymbol{s}_y), \boldsymbol{\phi}_Z(\boldsymbol{s}_z), \boldsymbol{\theta}_{\phi_Y}, \boldsymbol{\theta}_{\phi_B}) \cdot P(\boldsymbol{\phi}_Y(\boldsymbol{s}_y), \boldsymbol{\phi}_Z(\boldsymbol{s}_z), \boldsymbol{\theta}_{\phi_Y}, \boldsymbol{\theta}_{\phi_B})}{P(\boldsymbol{y}, \boldsymbol{z})} \tag{A12}$$

The first part of the numerator for the fraction can be broken down into:

$$P(\boldsymbol{y}, \boldsymbol{z} | \boldsymbol{\phi}_Y(\boldsymbol{s}_y), \boldsymbol{\phi}_Z(\boldsymbol{s}_z), \boldsymbol{\theta}_{\phi_Y}, \boldsymbol{\theta}_{\phi_B}) = P(\boldsymbol{y} | \boldsymbol{\phi}_Y(\boldsymbol{s}_y)) \cdot P(\boldsymbol{z} | \boldsymbol{\phi}_Z(\boldsymbol{s}_z)) \tag{A13}$$

The second part of the numerator for the fraction can be broken down into:

$$P(\boldsymbol{\phi}_Y(\boldsymbol{s}_y), \boldsymbol{\phi}_Z(\boldsymbol{s}_z), \boldsymbol{\theta}_{\phi_Y}, \boldsymbol{\theta}_{\phi_B}) = P(\boldsymbol{\phi}_Y(\boldsymbol{s}_y) | \boldsymbol{\phi}_Z(\boldsymbol{s}_z), \boldsymbol{\theta}_{\phi_Y}, \boldsymbol{\theta}_{\phi_B}) \cdot P(\boldsymbol{\phi}_Z(\boldsymbol{s}_z) | \boldsymbol{\theta}_{\phi_Y}, \boldsymbol{\theta}_{\phi_B}) \cdot P(\boldsymbol{\theta}_{\phi_Y}) \cdot P(\boldsymbol{\theta}_{\phi_B}) \tag{A14}$$

The above equations are inherently incorporated into the code implementation through the model definition using the Numpyro python package (Phan et al., 2019). The posterior distribution is approximated using MCMC, which returns realisations of $\boldsymbol{\phi}_Y(\boldsymbol{s}_y)$, $\boldsymbol{\phi}_Z(\boldsymbol{s}_z)$, $\boldsymbol{\theta}_{\phi_Y}$ and $\boldsymbol{\theta}_{\phi_B}$ from the posterior. The posterior predictive estimates of for example $\boldsymbol{\phi}_Y(\hat{\boldsymbol{s}})$ at any set of new locations $\hat{\boldsymbol{s}}$ across the domain is then given by the following:

$$P(\boldsymbol{\phi}_Y(\hat{\boldsymbol{s}}) | \boldsymbol{y}, \boldsymbol{z}) = \int P(\boldsymbol{\phi}_Y(\hat{\boldsymbol{s}}), \boldsymbol{\phi}_Y(\boldsymbol{s}_y), \boldsymbol{\phi}_Z(\boldsymbol{s}_z), \boldsymbol{\theta}_{\phi_Y}, \boldsymbol{\theta}_{\phi_B} | \boldsymbol{y}, \boldsymbol{z}) d\boldsymbol{\phi}_Y(\boldsymbol{s}_y) d\boldsymbol{\phi}_Y(\boldsymbol{s}_z) d\boldsymbol{\phi}_B(\boldsymbol{s}_z) d\boldsymbol{\theta}_{\phi_Y} d\boldsymbol{\theta}_{\phi_B} \tag{A15}$$

Where the integrand can be broken down into:

$$P(\boldsymbol{\phi}_Y(\hat{\boldsymbol{s}}), \boldsymbol{\phi}_Y(\boldsymbol{s}_y), \boldsymbol{\phi}_Z(\boldsymbol{s}_z), \boldsymbol{\theta}_{\phi_Y}, \boldsymbol{\theta}_{\phi_B} | \boldsymbol{y}, \boldsymbol{z}) = P(\boldsymbol{\phi}_Y(\hat{\boldsymbol{s}}) | \boldsymbol{\phi}_Y(\boldsymbol{s}_y), \boldsymbol{\phi}_Z(\boldsymbol{s}_z), \boldsymbol{\theta}_{\phi_Y}, \boldsymbol{\theta}_{\phi_B}) \cdot P(\boldsymbol{\phi}_Y(\boldsymbol{s}_y), \boldsymbol{\phi}_Z(\boldsymbol{s}_z), \boldsymbol{\theta}_{\phi_Y}, \boldsymbol{\theta}_{\phi_B} | \boldsymbol{y}, \boldsymbol{z})$$

$$\tag{A16}$$

The second part of this expression is equivalent to the posterior distribution defined earlier. The realisations from the posterior provided through the MCMC inference can be used as parameter values in the first part of the expression above to give a distribution that when sampled from provides posterior predictive realisations for $\boldsymbol{\phi}_Y(\hat{\boldsymbol{s}})$. In the case of Gaussian processes





the distribution of $P(\boldsymbol{\phi}_Y(\hat{\boldsymbol{s}})|\boldsymbol{\phi}_Y(\boldsymbol{s}_y),\boldsymbol{\phi}_Z(\boldsymbol{s}_z),\boldsymbol{\theta}_{\phi_Y},\boldsymbol{\theta}_{\phi_B})$ can be formulated in the following way, where to start take the joint distribution:

$$
\begin{bmatrix} \phi_Y(\hat{\boldsymbol{s}}) \\ \phi_Y(\boldsymbol{s}_y) \\ \phi_Z(\boldsymbol{s}_z) \end{bmatrix} \sim \mathcal{N}\left( \begin{bmatrix} m_{\phi_Y}(\hat{\boldsymbol{s}}) \\ m_{\phi_Y}(\boldsymbol{s}_y) \\ m_{\phi_Z}(\boldsymbol{s}_z) \end{bmatrix}, \begin{bmatrix} K_{\phi_Y}(\hat{\boldsymbol{s}},\hat{\boldsymbol{s}}) & K_{\phi_Y}(\hat{\boldsymbol{s}},\boldsymbol{s}_y) & K_{\phi_Y}(\hat{\boldsymbol{s}},\boldsymbol{s}_z) \\ K_{\phi_Y}(\boldsymbol{s}_y,\hat{\boldsymbol{s}}) & K_{\phi_Y}(\boldsymbol{s}_y,\boldsymbol{s}_y) & K_{\phi_Y}(\boldsymbol{s}_y,\boldsymbol{s}_z) \\ K_{\phi_Y}(\boldsymbol{s}_z,\hat{\boldsymbol{s}}) & K_{\phi_Y}(\boldsymbol{s}_z,\boldsymbol{s}_y) & K_{\phi_Z}(\boldsymbol{s}_z,\boldsymbol{s}_z) \end{bmatrix} \right) \tag{A17}
$$

Note, that since $\boldsymbol{\phi}_Y(s)$ and $\boldsymbol{\phi}_B(s)$ are independent and $\boldsymbol{\phi}_Z(s) = \boldsymbol{\phi}_Y(s) + \boldsymbol{\phi}_B(s)$, the covariance between the parameters $\boldsymbol{\phi}_Y(s)$ and $\boldsymbol{\phi}_Z(s)$ is simply $COV\left(\phi_Y(s),\phi_Z(s')\right) = COV\left(\phi_Y(s),\phi_Y(s')\right) = K_{\phi_Y}(s,s')$. Additionally, the mean and covariance terms for the process that generates $\boldsymbol{\phi}_Z(s)$ are computed as $m_{\phi_Z}(s) = m_{\phi_Y}(s) + m_{\phi_B}(s)$ and $K_{\phi_Z}(s,s') = K_{\phi_Y}(s,s') + K_{\phi_B}(s,s')$.

Then, defining the following:

$$
\quad U_1 = \begin{bmatrix} \phi_Y(\hat{\boldsymbol{s}}) \end{bmatrix}, U_2 = \begin{bmatrix} \phi_Y(\boldsymbol{s}_y) \\ \phi_Z(\boldsymbol{s}_z) \end{bmatrix}, \boldsymbol{U} = \begin{bmatrix} U_1 \\ U_2 \end{bmatrix}, M_1 = \begin{bmatrix} m_{\phi_Y}(\hat{\boldsymbol{s}}) \end{bmatrix}, M_2 = \begin{bmatrix} m_{\phi_Y}(\boldsymbol{s}_y) \\ m_{\phi_Z}(\boldsymbol{s}_z) \end{bmatrix}, \boldsymbol{M} = \begin{bmatrix} M_1 \\ M_2 \end{bmatrix} \tag{A18}
$$

$$
K_{11} = \begin{bmatrix} K_{\phi_Y}(\hat{\boldsymbol{s}},\hat{\boldsymbol{s}}) \end{bmatrix}, K_{12} = \begin{bmatrix} K_{\phi_Y}(\hat{\boldsymbol{s}},\boldsymbol{s}_y) & K_{\phi_Y}(\hat{\boldsymbol{s}},\boldsymbol{s}_z) \end{bmatrix} \tag{A19}
$$

$$
K_{21} = \begin{bmatrix} K_{\phi_Y}(\boldsymbol{s}_y,\hat{\boldsymbol{s}}) \\ K_{\phi_Y}(\boldsymbol{s}_z,\hat{\boldsymbol{s}}) \end{bmatrix}, K_{22} = \begin{bmatrix} K_{\phi_Y}(\boldsymbol{s}_y,\boldsymbol{s}_y) & K_{\phi_Y}(\boldsymbol{s}_y,\boldsymbol{s}_z) \\ K_{\phi_Y}(\boldsymbol{s}_z,\boldsymbol{s}_y) & K_{\phi_Z}(\boldsymbol{s}_z,\boldsymbol{s}_z) \end{bmatrix} \tag{A20}
$$

The distribution can be written as:

$$
\begin{bmatrix} U_1 \\ U_2 \end{bmatrix} \sim \mathcal{N}\left( \begin{bmatrix} M_1 \\ M_2 \end{bmatrix}, \begin{bmatrix} K_{11} & K_{12} \\ K_{21} & K_{22} \end{bmatrix} \right) \tag{A21}
$$

Where the conditional distribution $P(U_1|U_2)$ is well known for Gaussian distributions and is given as:

$$
P(U_1|U_2) = \mathcal{N}(M_{1|2}, K_{1|2}) \tag{A22}
$$

With parameters values:

$$
M_{1|2} = M_1 + K_{12}K_{22}^{-1}(U_2 - M_2) \tag{A23}
$$
$$
K_{1|2} = K_{11} - K_{12}K_{22}^{-1}K_{21} \tag{A24}
$$

This provides the distribution $P(U_1|U_2)$, which is equivalent to the distribution $P(\boldsymbol{\phi}_Y(\hat{\boldsymbol{s}})|\boldsymbol{\phi}_Y(\boldsymbol{s}_y),\boldsymbol{\phi}_Y(\boldsymbol{s}_z),\boldsymbol{\theta}_{\phi_Y})$ that is needed to compute the poserior predictive.





## A2 Non-hierarchical Case

In the non-hierarchical case used in Sect. 4.1, direct observations are assumed for $\phi_Y(s_y)$ and $\phi_Z(s_z)$. In this case the posterior for the model can be written out as:

$$P(\boldsymbol{\theta}_{\phi_Y},\boldsymbol{\theta}_{\phi_B}|\phi_Y(s_y),\phi_Z(s_z)) = \frac{P(\phi_Y(s_y),\phi_Z(s_z)|\boldsymbol{\theta}_{\phi_Y},\boldsymbol{\theta}_{\phi_B}) \cdot P(\boldsymbol{\theta}_{\phi_Y},\boldsymbol{\theta}_{\phi_B})}{P(\phi_Y(s_y),\phi_Z(s_z))} \tag{A25}$$


Where the first expression of the numerator can be broken down into:

$$P(\phi_Y(s_y),\phi_Z(s_z)|\boldsymbol{\theta}_{\phi_Y},\boldsymbol{\theta}_{\phi_B}) = P(\phi_Y(s_y)|\phi_Z(s_z),\boldsymbol{\theta}_{\phi_Y},\boldsymbol{\theta}_{\phi_B}) \cdot P(\phi_Z(s_z)|\boldsymbol{\theta}_{\phi_Y},\boldsymbol{\theta}_{\phi_B}) \tag{A26}$$

While the second part of the numerator can be split due to independence between the generating processes, such that:

$$P(\boldsymbol{\theta}_{\phi_Y},\boldsymbol{\theta}_{\phi_B}) = P(\boldsymbol{\theta}_{\phi_Y}) \cdot P(\boldsymbol{\theta}_{\phi_B}) \tag{A27}$$


As with the full hierarchical model, the above equations are inherently incorporated into the non-hierarchical code implementation, with the posterior distribution approximated using MCMC, which returns realisations of $\boldsymbol{\theta}_{\phi_Y}$ and $\boldsymbol{\theta}_{\phi_B}$ from the posterior. The posterior predictive estimates of for example $\phi_Y(\hat{s})$ at any set of new locations $\hat{s}$ across the domain is then given by the following:

$$P(\phi_Y(\hat{s})|\phi_Y(s_y),\phi_Z(s_z)) = \int P(\phi_Y(\hat{s}),\boldsymbol{\theta}_{\phi_Y},\boldsymbol{\theta}_{\phi_B}|\phi_Y(s_y),\phi_Z(s_z))d\boldsymbol{\theta}_{\phi_Y}d\boldsymbol{\theta}_{\phi_B} \tag{A28}$$


Where the integrand can be broken down into:

$$P(\phi_Y(\hat{s}),\boldsymbol{\theta}_{\phi_Y},\boldsymbol{\theta}_{\phi_B}|\phi_Y(s_y),\phi_Z(s_z)) = P(\phi_Y(\hat{s})|\boldsymbol{\theta}_{\phi_Y},\boldsymbol{\theta}_{\phi_B},\phi_Y(s_y),\phi_Z(s_z)) \cdot P(\boldsymbol{\theta}_{\phi_Y},\boldsymbol{\theta}_{\phi_B}|\phi_Y(s_y),\phi_Z(s_z)) \tag{A29}$$

The second part of this expression is equivalent to the posterior distribution defined earlier. The realisations from the posterior provided through the MCMC inference can be used as parameter values in the first part of the expression above to give a distribution that when sampled from provides posterior predictive realisations for $\phi_Y(\hat{s})$. The distribution in the first part of

the expression can be formulated in the same way as presented in Sect. A1.

## Appendix B: Data Generation

### B1 4.1

Define $\phi_Y$ as one parameter of the probability density function for the in situ observations and $\phi_Z$ as the corresponding parameter for the climate model output. The following relationship is then assumed $\phi_Z = \phi_Y + \phi_B$, where $\phi_B$ is the bias in the





parameter and is assumed independent of $\phi_Y$. The latent distributions that generate $\phi_Y$ and $\phi_B$ across the domain are assumed GPs with mean and covariance functions. The mean function is assumed constant for simplicity and the covariance function is taken as an RBF kernel with a kernel variance and length scale parameter (Eq. (5)). These hyper-parameters of the two latent generating processes are set for three scenarios, as given in Table 1.

For each scenario, a sample of the parameters $\phi_Y$ and $\phi_B$ is taken from the distributions $\mathcal{GP}_{\phi_Y}$ and $\mathcal{GP}_{\phi_B}$ at regularly spaced, high-resolution intervals. These samples are referred to here as complete realisations and represent underlying fields for each parameter across the domain, which the model aims to estimate. Direct 'observations' of the parameter $\phi_Y$ from the underlying field are simulated at lower-resolution, randomised locations after conditioning the distribution $\mathcal{GP}_{\phi_Y}$ on the complete realisation and introducing some noise. In order to simulate direct measurements of the parameter $\phi_Z$ of the climate model output, samples are first generated for $\phi_Y$ and $\phi_B$ at regularly spaced intervals after conditioning the distributions $\mathcal{GP}_{\phi_Y}$ and $\mathcal{GP}_{\phi_B}$ on the complete realisations, then the sum of these samples at each location is taken to give $\phi_Z$. The number of direct observations/measurements for each parameter under the different scenarios is given in Table 1.

### B2    4.2

A sample of the parameters $\mu_Y$, $\mu_B$, $\tilde{\sigma}_Y$ and $\tilde{\sigma}_B$ is taken from the distributions $\mathcal{GP}_{\mu_Y}$, $\mathcal{GP}_{\mu_B}$, $\mathcal{GP}_{\tilde{\sigma}_Y}$ and $\mathcal{GP}_{\tilde{\sigma}_B}$ at regularly spaced, high resolution intervals. These samples are referred to as complete realisations and represent the underlying fields for each parameter across the domain. A sample of the parameters for the in situ data ($\mu_Y$ and $\tilde{\sigma}_Y$) are also generated at a selection of lower-resolution, randomised locations after conditioning the latent distributions on the complete realisations. Observations of the in situ data $Y$ are then generated at these locations from the corresponding normal distribution. In the case of the climate model data $Z$, samples are first generated for $\mu_Y$, $\mu_B$, $\tilde{\sigma}_Y$ and $\tilde{\sigma}_B$ at regularly spaced intervals after conditioning the latent distributions on the complete realisations, then the sum of these samples at each location is taken to give $\mu_Z$ and $\tilde{\sigma}_Z$. The climate model data is then generated at these locations from the corresponding normal distribution. The number of locations and the number of samples per location are given in Table 4.

### Appendix C:  Prior and Posterior Distribution Examples

### C1    4.1

In Figure C1 the prior and posterior distributions are illustrated for each parameter given the measurement data in scenario one. As expected, it can be seen that the density of the posterior distribution for each parameter is concentrated closer to the value specified when generating the data than in the case of the prior distributions. The extent of the variation between the prior and posterior distributions depends on the specific parameter and the impact that parameter has on the likelihood of the measured data. As an example, the posterior distribution for the length scale of the latent GP that generates $\phi_Y$ across the domain (Fig. C1b) is more concentrated around the specified value compared to the equivalent length scale for the bias (Fig. C1e).





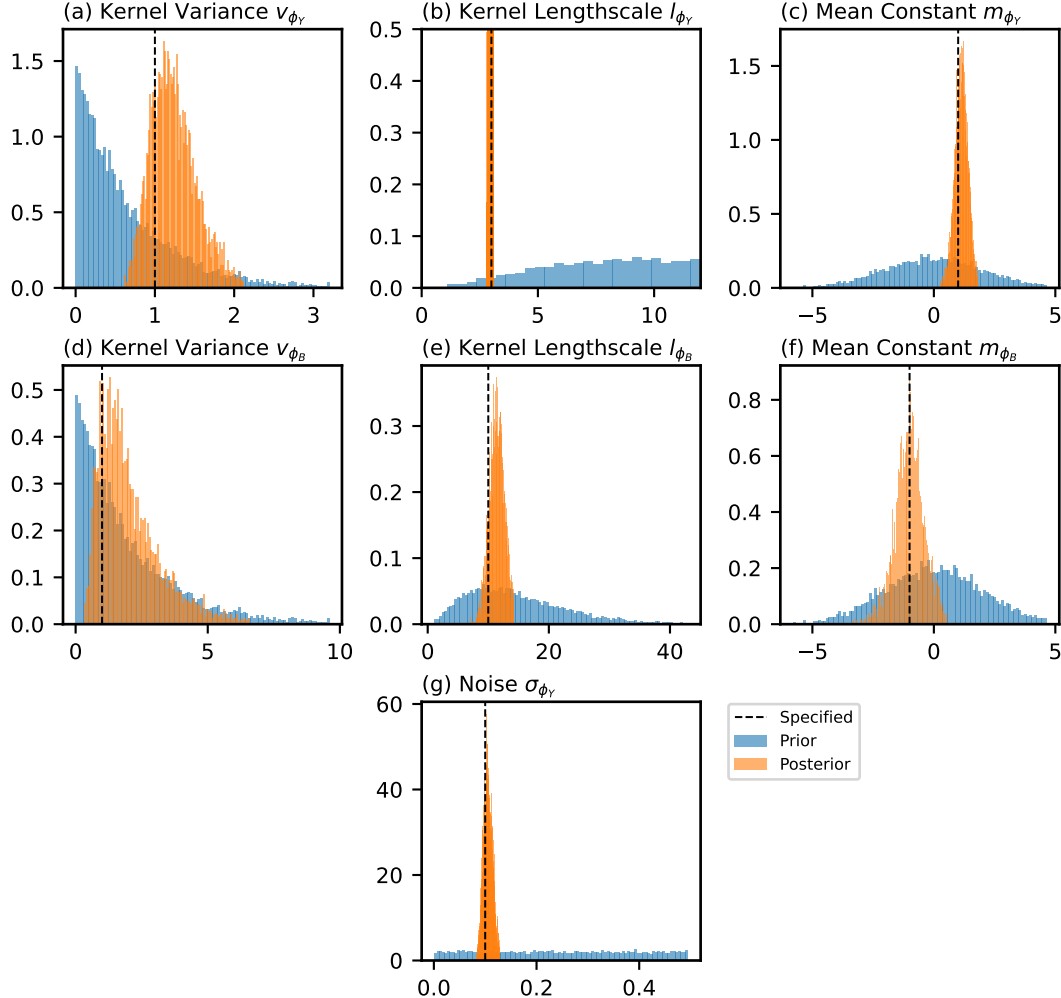

**Figure C1.** A figure illustrating the prior and posterior distributions for the parameters of the model in the case of scenario one. The value that was specified when generating the data is also shown.

## C2  4.2

The prior and posterior distributions for all parameters of the hierarchical model is presented in Figure C2. Inference on the parameters was performed using MCMC in a hierarchical Bayesian framework. Relatively non-informative prior distributions are chosen and are equivalent to the choice of priors in Sect. 4.1. The model assumes 4 generating GP distributions, one for each of the parameters $\mu_Y$, $\mu_B$, $\tilde{\sigma}_Y$ and $\tilde{\sigma}_B$. Each GP distribution has 3 associated parameters, being the mean constant $(m)$, the kernel variance $(v)$ and the kernel length scale $(l)$. The values specified in generating the data are shown as a dotted line for each parameter.



As expected, the posterior distribution for each parameter is concentrated closer to the value specified when generating the data than the relatively non-informative prior distributions. As with the non-hierarchical case, more confidence is shown on the value of some parameters than others in the posterior distributions, such as the posterior of the length scale of the latent GP
that generates $\mu_Y$ across the domain (Fig. C2b) compared to the equivalent length scale for that of the bias (Fig. C2h).

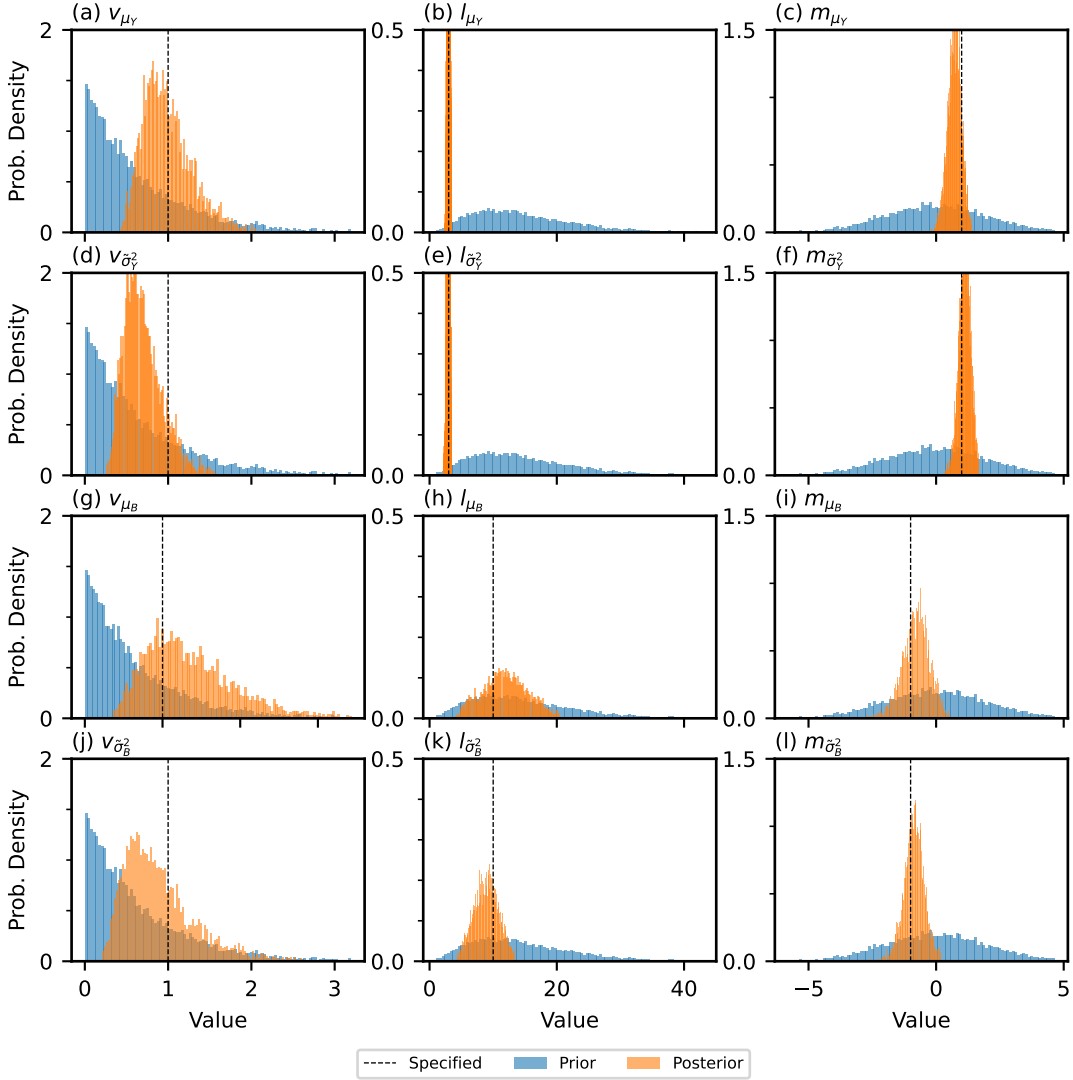

**Figure C2.** A figure illustrating the prior and posterior distributions for the parameters of the model in the case of the 1D hierarchical example. The value that was specified when generating the data is also shown.



*Author contributions.* J.Carter: Conceptualization, Methodology, Software, Validation, Formal analysis, Writing - Original Draft. E.Chacón-Montalván: Conceptualization, Methodology, Software, Validation, Formal analysis, Writing - Review & Editing, Supervision. A.Leeson: Conceptualization, Writing - Review & Editing, Supervision.

*Competing interests.* The authors declare that they have no conflict of interest.

*Acknowledgements.* J.Carter and A.Leeson are supported by the Data Science for the Natural Environment project (EPSRC grant number EP/R01860X/1). The code for analysis is written in Python 3.9.13 and makes extensive use the following libraries: Numpyro (Phan et al., 2019); TinyGP (Foreman-Mackey); NumPy (Harris et al., 2020); Matplotlib (Hunter, 2007).



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
