# Peer review of "Bias Correction of Climate Models using a Bayesian Hierarchical Model"

_EGUsphere, 2023_

## Author Comment (AC1)

**Authors' Response to Reviews of**

**Bias Correction of Climate Models using a Bayesian Hierarchical Model**

J. Carter
*Geoscientific Model Development,*
* * *
**RC:** *Reviewers' Comment*,  AR: Authors' Response,  ☐ Manuscript Text

Thank you for the review of the manuscript. We are very grateful for your careful and insightful comments, which have contributed to the improvement of the original manuscript. We have worked hard to incorporate the feedback into the revised manuscript and have detailed here our thoughts and any changes made for each comment individually below. We hope you find the response and changes satisfactory.

A revised manuscript has also been provided, along with a LaTex-diff document highlighting changes. Due to the re-structuring of the manuscript the LaTex-diff document is difficult to follow, so a brief summary of the main changes is described here:

1. The structure has been updated to a more conventional format to align with the reviewers comments. To make this work some sections have been moved to the appendix.

2. All sections have been partially re-written to align with the new structure and to further improve clarity on some points for the reader. Some additional paragraphs are included to align with specific reviewer's comments

3. Figures and Tables have been partially adjusted to further improve clarity and consistency with the re-written text.

**1. Referee 1**

**1.1. Concerns**

**1.1.1**

RC:

*1. The experiments and results are only on simulated data. While the experiments are beneficial in understanding how the model would perform under different data distributions, the benefits of the framework on real observed / climate model data have not been validated.*

AR:

It's agreed that applying the framework to simulated examples only doesn't validate the effectiveness of the methodology against real-world datasets. However, the aim of this paper, which is already ambitious is to provide an initial proof of concept against three simplified and varied simulated case studies and with the expectation that further papers will validate the methodology against real-world case studies. Extending the current paper to include a real-world scenario would involve extensive additional work, such as intricate data cleaning, processing and exploration of the variables under study, as well as the expansion of our methodology to incorporate mean functions dependent on relevant spatial predictors like elevation and finally writing around the specific importance of results to the field of study. This would add significant added complexity to an already complex manuscript and we believe it is better suited to inclusion in a separate paper. It is believed including this in the current paper would over-complicate an already long paper and so instead we have made adjustments to the abstract and discussion to make it clear that future pieces of work validating against real-world applications are important:

Relevant adjustment to abstract:

> This paper focuses on one-dimensional simulated examples for clarity, although the code implementation is developed to also work on multi-dimensional input data, encouraging follow-on real-world application studies that will further validate performance and remaining limitations. The Bayesian framework supports uncertainty propagation under model adaptations required for specific applications, providing a flexible approach that increases the scope to data assimilation tasks more generally.

Relevant adjustments to discussion:

> The simulated examples presented provide an initial proof of concept, although future studies validating the methodology against real-world applications are important for understanding the remaining limitations and areas for further development. The current primary limitation is expected to be that the underlying spatial covariance structures are assumed stationary. That is that the covariance length scale is assumed constant across the domain, whereas for real-world applications over large and complex topographic domains the length scale will be expected to change depending on the specific topography of the region. Further development of the methodology to incorporate non-stationary kernels would therefore be valuable, although is beyond the scope of this paper. Another important limitation to consider is the assumption that the bias is time independent. In situations where the bias varies gradually through time and uniformly across the domain, the methodology can be further developed such that the mean function of the GPs is modelled with a time dependency. If the bias varies in time non-uniformly across the domain, spatiotemporal GPs will need to be considered, which is again beyond the scope of this paper. Secondary limitations, include the assumption that the unbiased and biased components of the PDF parameter values are independent. In situations where there is a dependence between these components, the methodology presented is still expected to perform adequately, although information is lost by not modelling the dependency explicitly. Additionally, many real-world applications will necessitate specific

model adjustments, such as incorporating a mean function dependent on factors like elevation and latitude. Finally, the computational complexity of the model is an important remaining consideration, with inference time of GPs scaling as the cube of the number of data points. Incorporating techniques from the literature such as using sparse variational GPs (SVGP) (Hensman et al., 2015), nearest-neighbor GPs (NNGP) (Datta et al., 2016) or upscaling the climate model output, while outside the scope of this paper, will aid computational performance under demanding real-world scenarios and will facilitate further model development.

Additionally, it is recognised that the title of the paper is potentially misleading with the focus being on the application and so adjustment is made from 'Bias Correction of Climate Models using a Bayesian Hierarchical Model' to 'Bayesian Hierarchical Model for Bias Correcting Climate Models'. This hopefully puts more focus on the model development.

**1.1.2**

**RC:**

*2. The clarity of the paper can be improved. Addressing these concerns may make it easier to follow the discussion in the paper.*

- *In several places, the term "model" can be clarified. For instance, if it is the climate model or the GP model that is being discussed.*

- *Phrases like "spatially varying parameter of the PDF for the climate model output" are a little unclear. Does this refer to the hyper-parameters from the GP model?*

AR:

We thank the reviewer for this suggestion and have edited the manuscript accordingly. Many small adjustments have been made to sentences that could be perceived as ambiguous and in particular attention has been made to always include context around the term 'model' for clarity. For example, the full terms 'climate model', 'hierarchical model' and 'shared process model' are all used rather than just using 'model' to make it clear what is being referred to.

Phrases like 'spatially varying parameter of the PDF for the climate model output' have been shortened to just 'parameter of the PDF for the climate model' to improve clarity. For example, if at each site the PDF of the data is considered normal then this gives $Z(s_i) \sim \mathcal{N}(\mu(s_i), \sigma(s_i))$, where either of the parameters $\mu(s_i)$ and $\sigma(s_i)$ are what we are referring to in the previous sentence. When we are referring to the parameters of the latent GPs we've made sure to always use the term 'hyper-parameters' for consistency.

**1.1.3**

**RC:**

*3. Figures 3 and 5 have not been discussed as part of the main results/discussion. It is unclear how these figures factor into the main discussion.*

AR:

Figures 3 and 5 were part of the methodology section. Figure 3a is intended to help the reader visualise how the covariance decays with distance for a given length scale and 3b to visualise how conditioning on data impacts the GP distribution, see lines 261-262 of original manuscript: 'In Fig. 3, an example of how the covariance decays with distance is given for the RBF kernel and realisations of a conditioned GP with the equivalent kernel are illustrated.' Figure 5 has now been removed but was originally intended to just provide an illustration of the nature of realisations from GPs.

To further improve clarity to the reader a more convention structure has been adopted with a single 'Results' section and with the subsection 3.3 'Capturing Spatial Structure with Gaussian Processes' (that contains Fig. 3 and 5 and was part of the methodology) now moved to the appendix to improve the flow of the main part of the paper. Figure 3 is now Fig. B1 and Fig. 5 is removed since it is unnecessary with the new 'Data Generation' section of the revised manuscript.

**1.1.4**

RC:

*4. It is unclear if the authors are separating the train and test set to evaluate the model performance for the GP models. Training on test set may result in overfitting and high R2 scores.*

AR:

We thank the reviewer for identifying this area for further clarification and have edited the manuscript accordingly. A specific section on 'Data Generation' for the simulated examples has been added to the paper (Sect. 3), where the train and test sets are discussed. The key point is to recognise that the unbiased parameters are observed at locations $s_y$, denoted as $\phi_Y(s_y)$, while the biased parameters are observed at locations $s_z$, denoted as $\phi_Z(s_z)$; both comprise the training set. The objective is to assess how effectively we can recover the unbiased parameters at locations where the biased parameters were observed, denoted as $\phi_Y(s_z)$, which constitutes the testing set. The distinction from conventional training and testing sets arises from having different sources of information, each with distinct sampling units.

For the non-hierarchical examples, where $R^2$ scores are computed, the following paragraph describes the training and test sets:

> For each scenario, a sample of the parameters $\phi_Y(s^\star)$ and $\phi_B(s^\star)$ is taken from the distributions $\mathcal{GP}_{\phi_Y}$ and $\mathcal{GP}_{\phi_B}$ at regularly spaced, high-resolution intervals. These samples are referred to here as complete realisations and represent underlying fields for each parameter across the domain. The complete realisations of $\phi_Y(s^\star)$ are sampled at lower-resolution, randomised locations, with the addition of some noise, to provide direct simulated 'in situ observations' of the parameter $\phi_Y(s_y)$. In order to simulate input data for the parameter $\phi_Z(s_z)$ of the climate model output, the complete realisations of $\phi_Y(s^\star)$ and $\phi_B(s^\star)$ are sampled at regularly spaced intervals to provide $\phi_Y(s_z)$ and $\phi_B(s_z)$, then the sum of these samples at each location is taken to give $\phi_Z(s_z)$. The input data for inference is then $\phi_Y(s_y)$ and $\phi_Z(s_z)$ and can be considered as the training set, while the underlying realisations generated for $\phi_Y(s_z)$ are the test set used for validating the model performance.

Additionally, in the results section, some writing is included to explain how the $R^2$ score is computed:

> The relative performance of the shared and single process models is quantified by computing $R^2$ scores between the predictions of $\phi_Y(s_z)$ and the actual values used in generating the data (although not used in training), with results presented in Table 4.

That is the model performance is evaluated at the climate model locations ($s_z$) but for estimates of the unbiased PDF parameter $\phi_Y(s_z)$, which is not the same as the input data $\phi_Y(s_y)$ and $\phi_Z(s_z)$. It is logical to measure performance by estimates of the unbiased parameter at the simulated climate model locations, since this is what is needed for bias correction.

**1.1.5**

RC:

*5. The abstract and introduction talk about several potential benefits of the model. In the current structuring, it is a little*

*difficult to follow along which statements correspond with validated claims in the paper and which statements are just potential benefits / future directions. More emphasis on statements that support the results presented in the result section may strengthen the case for the proposed framework.*

AR:

It is agreed that more emphasis on statements supported directly by the results is desirable. The abstract has been restructured and re-written in places to hopefully reflect this, in particular see this part:

> This paper proposes a novel Bayesian bias correction framework that propagates uncertainty robustly and models underlying spatial covariance patterns. Shared latent Gaussian processes are assumed between the in situ observations and climate model output with the aim of partially preserving the covariance structure from the climate model after bias correction, which is based on well-established physical laws. Results demonstrate added value in modelling shared generating processes under several simulated scenarios, with most value added for the case of sparse in situ observations and smooth underlying bias. Additionally, the propagation of uncertainty to a simulated final bias corrected time series is illustrated, which is of key importance to a range of stakeholders, from climate scientists engaged in impact studies, decision makers trying to understand the likelihood of particular scenarios and individuals involved in climate change adaption strategies where accurate risk assessment is required for optimal resource allocation.

The robust uncertainty propagation and modelling of spatial covariance patterns are inherent properties of the model. Results illustrate these properties as well as focusing on the added value provided by considering shared latent GPs between in situ observations and the climate model output.

**1.1.6**

RC:

*6. Similarly, in the discussion section, it may be helpful to separate the potential directions / future work in a separate paragraph.*

AR:

It's agreed that having a separate paragraph where current limitations and potential future directions is discussed would be useful for the reader. This has now been incorporated into the discussion section, where the following paragraph has been added:

> The simulated examples presented provide an initial proof of concept, although future studies validating the methodology against real-world applications are important for understanding the remaining limitations and areas for further development. The current primary limitation is expected to be that the underlying spatial covariance structures are assumed stationary. That is that the covariance length scale is assumed constant across the domain, whereas for real-world applications over large and complex topographic domains the length scale will be expected to change depending on the specific topography of the region. Further development of the methodology to incorporate non-stationary kernels would therefore be valuable, although is beyond the scope of this paper. Another important limitation to consider is the assumption that the bias is time independent. In situations where the bias varies gradually through time and uniformly across the domain, the methodology can be further developed such that the mean function of the GPs is modelled with a time dependency. If the bias varies in time non-uniformly across the domain, spatiotemporal GPs will need to be considered, which is again beyond the scope of this paper. Secondary limitations, include the assumption that the unbiased and biased components of the PDF parameter values are independent. In situations where there is a dependence

between these components, the methodology presented is still expected to perform adequately, although information is lost by not modelling the dependency explicitly. Additionally, many real-world applications will necessitate specific model adjustments, such as incorporating a mean function dependent on factors like elevation and latitude. Finally, the computational complexity of the model is an important remaining consideration, with inference time of GPs scaling as the cube of the number of data points. Incorporating techniques from the literature such as using sparse variational GPs (SVGP) (Hensman et al., 2015), nearest-neighbor GPs (NNGP) (Datta et al., 2016) or upscaling the climate model output, while outside the scope of this paper, will aid computational performance under demanding real-world scenarios and will facilitate further model development.

**1.2. Questions**

**1.2.1**

**RC:**

*1. Why is the uncertainty not being plotted for the phi z case in Figures 7 and 9?*

AR:

It is correct that for Fig. 7 & 10 (numbering before revision, where the posterior predictive distributions are plotted) we do not currently include predictions on $\phi_Z$, although we could. The reason for this is because the purpose of the hierarchical model is primarily to split up the biased signal $\phi_Z(s_z)$ into it's unbiased $\phi_Y(s_z)$ and biased components $\phi_B(s_z)$. To make this more clear in the figure it is deemed best to only show these predictions. Estimating $\phi_Z(s_z)$ itself is simple due to the high temporal coverage of climate model output and so predictions would line up closely to the true values with low uncertainty, not being of much interest to the reader.

**1.2.2**

**RC:**

*2. Is the model performance evaluated on the train set itself?*

AR:

The training dataset and the test dataset are different. In the non-hierarchical examples the input data are the PDF parameter values of $\phi_Y(s_y)$ and $\phi_Z(s_z)$. Performance is evaluated against posterior predictive estimates of the unbiased PDF parameter values at the locations of the simulated climate model output $\phi_Y(s_z)$. A section has been added to the main paper on this for further clarity to the reader, see AR: 1.1.4.

**1.2.3**

**RC:**

*3. How is the "robustness" of uncertainty propagation being evaluated?*

AR:

The statement that having a Bayesian framework allows robust uncertainty propagation refers to how predictions inherently capture uncertainty in each component of the framework in a mathematically justified way. Meaning that any adjustments to the methodology, such as including a mean function in the GPs, can be made while inherently retaining mathematically justified uncertainty propagation. The uncertainty estimates in predictions of the unbiased PDF parameter values at the climate model locations $\phi_Y(s_z)$ are evaluated as reasonable by comparison to the true underlying values, which are computed as part of the data generation.

**1.2.4**

**RC:**

*4. There may be concerns relating to the computational complexity of the framework. Since the three terms are modelled as exact GPs, the time and space complexity will grow quickly for larger datasets. Are there any variants/modifications to the framework that are being studied to overcome these challenges?*

AR:

It's correct that the computational complexity of GPs scale as the cube of the number of data points, so the time for inference of the model's parameters quickly becomes an important consideration when extended to real-world scenarios. A section has been added on this at the end of the discussion:

> Finally, the computational complexity of the model is an important remaining consideration, with inference time of GPs scaling as the cube of the number of data points. Incorporating techniques from the literature such as using sparse variational GPs (SVGP) (Hensman et al., 2015), nearest-neighbor GPs (NNGP) (Datta et al., 2016) or upscaling the climate model output, while outside the scope of this paper, will aid computational performance under demanding real-world scenarios and will facilitate further model development.

As mentioned there are various methods in the literature for improving the computational cost of GPs, such as SVGPs (Hensman et al., 2015) and NNGPs (Datta et al., 2016). While implementing these is outside the scope of this initial paper and would add significant complexity, it's an area for future development.

**1.2.5**

**RC:**

*5. Since simulated data is being used, are the model performances averaged over several repetitions?*

AR:

In the paper the model performance is not evaluated over multiple repetitions of simulated data for each scenario. This is not done as it is believed it would not add much to the findings and discussion. The findings for each of the case studies makes logical sense and the studies are not intended to provide a comprehensive overview of performance, just an initial proof of concept. It is agreed that adding repetitions would improve the accuracy of $R^2$ scores for each scenario but it is believed this wouldn't have much effect on interpretation.

**1.2.6**

**RC:**

*6. Does the proposed framework use quantile mapping along with GP modeling for bias correction?*

AR:

This is correct, after making predictions on the unbiased PDF parameter values at the climate model output locations $\phi_Y(\boldsymbol{s}_z)$, quantile mapping is then used to apply the final bias correction to the climate data. To make this more clear to the reader a figure has been included in the methodology showing the key steps of the framework proposed:

[Figure]

**Figure 2.** The full bias correction framework proposed in this paper broken down into the key steps.

**1.2.7**

RC:

*7. How many iterations were used for the MCMC sampling?*

AR:

For the MCMC sampling 1000 iterations were used for warm-up and then 2000 samples taken. This was found to be adequate for convergence. Some writing has been added to the results section to reflect this.

Inference is done in a Bayesian framework using MCMC and the No-U-Turn Sampler (NUTS) algorithm (Hoffman and Gelman, 2014) implemented in Numpyro (Phan et al., 2019). For the MCMC sampling 1000 iterations were used for warm-up and then 2000 samples taken, which was found to be adequate for convergence.

**1.3.  Other Comments:**

**1.3.1**

RC:

*1. The paper can benefit from more discussion on how uncertainty estimates are impacted by bias correction.*

AR:

Regional climate model output for a variable over a region often doesn't include an uncertainty (Carter et al., 2022). In this paper bias correction is applied considering output from a single climate model simulation without any estimated uncertainty. In the bias correction methodology proposed, estimates of the unbiased parameters of the PDF, derived from in situ observations and the climate model output itself are made and it is the uncertainty in the estimates of the unbiased PDF parameters that is captured and propagated to the resulting bias corrected time series.

Considering cases where uncertainty is provided in the original RCM output, through for example running multiple climate model simulations and creating an ensemble, is outside the scope of this initial piece of work. Future studies could extend the methodology to address this by for example averaging the $n_k$ ensemble members and introducing an error term, such that
$$\bar{\boldsymbol{z}}_{s_i} = \frac{\sum_{k=1}^{n_k} \boldsymbol{z}_{k,s_i}}{n_k} + \boldsymbol{e}_{s_i}.$$

**1.3.2**

RC:

*2. It is unclear to what extent the quantile mapping mitigates the bias and to what extent the proposed GP framework is useful in bias correction.*

AR:

The framework proposed for bias correction in this paper consists of the following steps:

1. Apply MCMC inference on the hierarchical model to obtain estimates of the PDF parameters $\phi_Y(s_y)$ and $\phi_Z(s_z)$, as well as estimates of the hyper-parameters $\theta_Y$ and $\theta_B$ for the generating spatial processes (GPs).

2. Sample from the posterior predictive distribution $P(\phi_Y(s_z)|y, z)$ to get estimates of the unbiased PDF parameters at the climate model grid cells.

3. Use the samples of $\phi_Y(s_z)$ and $\phi_Z(s_z)$ to apply quantile mapping to the climate model time series at each grid cell, resulting in a bias corrected output with uncertainty bands.

Quantile mapping is only applied in step 3 after being able to predict, with uncertainty, the unbiased parameters $\phi_Y(\cdot)$ at location $s_z$. Quantile mapping is applied on the climate model output to match the PDF of the data onto that of the estimated unbiased PDF. The quantiles of the climate model data are remapped onto the quantiles of the observed data, thus bias correcting the output. The GPs allow estimates of the unbiased parameters to be made at the climate model output locations while considering the underlying spatial covariance between points, providing a natural and flexible way to consider the spatial structure in the parameters. To make this overall framework more clear, the methodology section has been re-written and includes discussion of quantile mapping (see below) along with a figure showing all the key steps of the framework (see AR: 1.2.6).

> After obtaining multiple realisations of $\phi_Y(s_z)$ and $\phi_Z(s_z)$ quantile mapping is then used to bias correct the climate model time series at every grid cell location. Specifically, for each value of the time series from the climate model output at a given point ($z_{s'_i,j}$), this involves finding the percentile of that value using the parameters $\phi_Z(s'_i)$ and then mapping the value onto the corresponding value of the equivalent percentile of the PDF estimated for the unbiased process, defined through the parameters $\phi_Y(s'_i)$. The cumulative density function (CDF) returns the percentile of a given value and the inverse CDF returns the value corresponding to a given percentile, which results in the following correction function $\hat{z}_{s_i,j} = F_{Y_{s_i}}^{-1}(F_{Z_{s_i}}(z_{s_i,j}))$, where $F$ represents the CDF at a specific site. The CDF can be estimated as an integral over the parametric form assumed for the PDF. The Bayesian hierarchical model presented provides a collection of realisations for $\phi_Y(s_z)$ and $\phi_Z(s_z)$ from an underlying latent distribution. Applying quantile mapping with each set of realisations then results in a collection of bias corrected time series, with an expectation and uncertainty.

**1.3.3**

RC:

*3. In the abstract, it is suggested that the model provides value addition that is more than that of "alternative approaches." A comparative analysis with the alternative approaches may help substantiate this claim.*

AR:

It's agreed this is important and a comparison is given for the non-hierarchical examples (Sect. 4.1 of updated manuscript) against the most relevant similar approach identified in the literature (Lima et al., 2021). An appendix section 'Shared and Single Process Model Comparison' has been included to illustrate the difference in the two approaches.

**Appendix E: Shared and Single Process Model Comparison**

550 In the main paper comparisons are made made between the shared process and single process models. The shared process model is the hierarchical model proposed in this paper, while the single process model represents a similar approach taken from the literature, see Lima et al. (2021). The two models are shown in Fig. E1. In both models the random variables for the in-situ observations $Y(s)$ and climate model output $Z(s)$ have PDFs with the collection of parameters $\phi_Y(s)$ and $\phi_Z(s)$ respectively. In the case of the shared process model, $\phi_Z(s)$ is modelled as the sum of $\phi_Y(s)$ and some independent bias $\phi_B(s)$.

555 The parameters $\phi_Y(s)$ and the corresponding bias $\phi_B(s)$ are each themselves modelled over the domain as generated from Gaussian processes with hyper-parameters $\theta_Y$ and $\theta_B$. The unbiased parameters $\phi_Y(s)$ and hyper-parameters $\theta_Y$ are inferred from both the in situ data and climate model output. Posterior predictive estimates of $\phi_Y(s_z)$ are made by conditioning on both sets of data. In the case of the single process model, the PDF parameters for the climate model output and in situ observations are treated as independent. Only one latent GP is considered with the unbiased parameters $\phi_Y(s)$ and hyper-parameters $\theta_Y$

560 inferred from in situ observations alone. Posterior predictive estimates of $\phi_Y(s_z)$ are also made by conditioning on just in situ observations.

[Figure]

**Figure E1.** Plate diagram illustrating the difference between the shared process hierarchical model presented in this paper and the single process model that comparisons are made against.

**1.3.4**

**RC:**

*4. Appendix equations A1 to A11 were really helpful and can potentially be included in the main section*

AR:

It's good that this section was helpful in explaining the methodology. We've re-written the methodology to include elements of this section (A1-A11) as suggested. The posterior and posterior predictive formulations (A12-A29) are kept in the appendix to keep the main part of the paper more approachable for researchers without an expertise in statistics.

**2. Referee 2**

**2.1. Main Comments**

**2.1.1**

**RC:**

*1. Manuscript Structure: The reviewer recommends a comprehensive revision of the submitted manuscript to adhere to a conventional structure, encompassing introduction, data, methodology, results, discussion, and conclusion. The current manuscript structure leads to repetitive content, with similar discussions occurring in sections such as 2.3 and later in section 3, particularly regarding the limitations of pre-existing bias correction approaches. To improve continuity, the reviewer suggests maintaining focus within each section, as the current nested content creates a discontinuous logic flow that may be challenging for readers to follow.*

AR:

We thank the reviewer for their constructive comment and have edited the manuscript accordingly. The paper has undergone a comprehensive restructuring, adhering to a more conventional structure as recommended, and sections have been re-written to remove repetitive content and to improve the flow. The old and new structures are shown below:

Old structure:

1. Introduction
2. Bias in Climate Models
   - 2.1 Bias in Random Variables
   - 2.2 Bias with Spatially Varying Parameters
   - 2.3 Bias Correction
3. Bias Prediction Methodology
   - 3.1 Model Overview
   - 3.2 Specific Model Example
   - 3.3 Capturing Spatial Structure with Gaussian Processes
4. Simulated Examples
   - 4.1 Shared Latent Generating Processes: Non Hierarchical Example
     - 4.1.1 Data Generation
     - 4.1.2 Results
   - 4.2 Bayesian Framework: Hierarchical Example
     - 4.2.1 Data Generation
     - 4.2.2 Results
5. Discussion
6. Conclusion
7. Appendix
   - 7.1 Posterior and Posterior Predictives
   - 7.2 Data Generation
   - 7.3 Prior and Posterior Distribution Examples

New structure:

1. Introduction
2. Methodology
3. Data Generation
   - 3.1 Non Hierarchical Examples: Data Generation
   - 3.2 Hierarchical Example: Data Generation
4. Results
   - 4.1 Non Hierarchical Examples: Results
   - 4.2 Hierarchical Example: Results
5. Discussion
6. Conclusion
7. Appendix
   - 7.1 Bias in Climate Models
   - 7.2 Capturing Spatial Structure with Gaussian Processes
   - 7.3 Posterior and Posterior Predictive Formulation
   - 7.4 Specific Example with Temperature
   - 7.5 Shared and Single Process Model Comparison
   - 7.6 Prior and Posterior Hyper-Parameter Distributions

**2.1.2**

**RC:**

*2. Novelty: The reviewer observes that the novelty of the submitted manuscript is not adequately emphasized. In the submitted manuscript, only a handful of bias correction related work were mentioned in the introduction section. However, none of the cited works applied the Bayesian-related technique for bias correction. The reviewer believes that there should be more literature available that are related to this study. Notably, the authors themselves acknowledge the foundation of the submitted work on Lima et al. (2021) in section 3. However, the reviewer is left questioning the relationship between Lima et al. (2021) and the current study. To mitigate potential confusion and provide a more comprehensive background, the reviewer recommends introducing this information more explicitly. This comment echoes concerns outlined in detail in Comment #1.*

AR:

It's agreed that it's important to make comparisons to other techniques, such as Lima et al. (2021), clear to the reader from an early point. This is done in section 2.3 (old format) but in alignment with the referees suggestion the paper has been restructured and the information instead included in the introduction to hopefully emphasise the novelty adequately.

> The approach presented builds on that of Lima et al., 2021, which models the in situ observational data as generated from a GP and uses quantile mapping (Qian and Chang, 2021) to apply the correction to the climate model output. In Lima et al., 2021 the spatial covariance structure of the climate model output is not considered and uncertainty is not propagated to the final bias corrected time series. The novelty of the approach proposed here is that shared latent GPs are modelled between the climate model output and the in situ observational data, which aims to incorporate information from the physically realistic spatial patterns of the climate model output in predictions of the unbiased field. Additionally, uncertainty is propagated through the quantile mapping step, which results in uncertainty bands on the bias corrected output. The approach is developed with the focus of applying bias correction to regions with sparse in situ observations, such as over Antarctica, where capturing uncertainty in the correction is of key importance and where including data from all sources during inference is particularly valuable. Performance under simulated scenarios with differing data density and underlying covariance length scales is evaluated in this paper and the potential added value assessed when compared with the approach in Lima et al., 2021.

Additionally, a section has been added to the appendix, which the reader is referred to further clarify the difference between the approach suggested and that applied in Lima et al., 2021, see below.

**Appendix E: Shared and Single Process Model Comparison**

550    In the main paper comparisons are made made between the shared process and single process models. The shared process model is the hierarchical model proposed in this paper, while the single process model represents a similar approach taken from the literature, see Lima et al. (2021). The two models are shown in Fig. E1. In both models the random variables for the in-situ observations $Y(s)$ and climate model output $Z(s)$ have PDFs with the collection of parameters $\phi_Y(s)$ and $\phi_Z(s)$ respectively. In the case of the shared process model, $\phi_Z(s)$ is modelled as the sum of $\phi_Y(s)$ and some independent bias $\phi_B(s)$.

555    The parameters $\phi_Y(s)$ and the corresponding bias $\phi_B(s)$ are each themselves modelled over the domain as generated from Gaussian processes with hyper-parameters $\boldsymbol{\theta}_Y$ and $\boldsymbol{\theta}_B$. The unbiased parameters $\phi_Y(s)$ and hyper-parameters $\boldsymbol{\theta}_Y$ are inferred from both the in situ data and climate model output. Posterior predictive estimates of $\phi_Y(\boldsymbol{s}_z)$ are made by conditioning on both sets of data. In the case of the single process model, the PDF parameters for the climate model output and in situ observations are treated as independent. Only one latent GP is considered with the unbiased parameters $\phi_Y(s)$ and hyper-parameters $\boldsymbol{\theta}_Y$

560    inferred from in situ observations alone. Posterior predictive estimates of $\phi_Y(\boldsymbol{s}_z)$ are also made by conditioning on just in situ observations.

[Figure]

**Figure E1.** Plate diagram illustrating the difference between the shared process hierarchical model presented in this paper and the single process model that comparisons are made against.

**2.1.3**

**RC:**

*3. Experiment design: Asides from comment#1 where it is suggested the author re-organize the manuscript, the reviewer suggests conducting experiments with the proposed technique using real climate model output. Experiments on climate model output could better highlight the strength and further verify the effectiveness of the proposed techniques.*

AR:

It is agreed that applying the proposed framework to a real-world case study would provide valuable insights into the strengths and limitations of the proposed methodology, although this is considered outside the scope of the paper, which provides an initial proof of concept. As mentioned in response AR:1.1.1, the aim of this paper, which is already ambitious is to provide an initial proof of concept against three simplified and varied simulated case studies and with the expectation that further papers will validate the methodology against real-world case studies. Extending the current paper to include a real-world scenario

would involve extensive additional work, such as intricate data cleaning, processing and exploration of the variables under study, as well as the expansion of our methodology to incorporate mean functions dependent on relevant spatial predictors like elevation and finally writing around the specific importance of results to the field of study. This would add significant added complexity to an already complex manuscript and we believe it is better suited to inclusion in a separate paper. It is believed including this in the current paper would over-complicate an already long paper and so instead we have made adjustments to the abstract and discussion to make it clear that future pieces of work validating against real-world applications are important:

Relevant adjustment to abstract:

> This paper focuses on one-dimensional simulated examples for clarity, although the code implementation is developed to also work on multi-dimensional input data, encouraging follow-on real-world application studies that will further validate performance and remaining limitations. The Bayesian framework supports uncertainty propagation under model adaptations required for specific applications, providing a flexible approach that increases the scope to data assimilation tasks more generally.

Relevant adjustments to discussion:

> The simulated examples presented provide an initial proof of concept, although future studies validating the methodology against real-world applications are important for understanding the remaining limitations and areas for further development. The current primary limitation is expected to be that the underlying spatial covariance structures are assumed stationary. That is that the covariance length scale is assumed constant across the domain, whereas for real-world applications over large and complex topographic domains the length scale will be expected to change depending on the specific topography of the region. Further development of the methodology to incorporate non-stationary kernels would therefore be valuable, although is beyond the scope of this paper. Another important limitation to consider is the assumption that the bias is time independent. In situations where the bias varies gradually through time and uniformly across the domain, the methodology can be further developed such that the mean function of the GPs is modelled with a time dependency. If the bias varies in time non-uniformly across the domain, spatiotemporal GPs will need to be considered, which is again beyond the scope of this paper. Secondary limitations, include the assumption that the unbiased and biased components of the PDF parameter values are independent. In situations where there is a dependence between these components, the methodology presented is still expected to perform adequately, although information is lost by not modelling the dependency explicitly. Additionally, many real-world applications will necessitate specific model adjustments, such as incorporating a mean function dependent on factors like elevation and latitude. Finally, the computational complexity of the model is an important remaining consideration, with inference time of GPs scaling as the cube of the number of data points. Incorporating techniques from the literature such as using sparse variational GPs (SVGP) (Hensman et al., 2015), nearest-neighbor GPs (NNGP) (Datta et al., 2016) or upscaling the climate model output, while outside the scope of this paper, will aid computational performance under demanding real-world scenarios and will facilitate further model development.

Additionally, as also mentioned in response AR:1.1.1 it is recognised that the title of the paper is potentially misleading with the focus being on the application and so adjustment is made from 'Bias Correction of Climate Models using a Bayesian Hierarchical Model' to 'Bayesian Hierarchical Model for Bias Correcting Climate Models'. This hopefully puts more focus on the model development.

**2.1.4**

**RC:**
*4. Methodology: It appears to the reviewer that the methodology of the submitted study should be better introduced and*

*clearly explained. It appears to the reviewer that important information is missing which prevents the audience from following and telling the scientific and technical values of the proposed bias correction approach. For instance, to the reviewer's understanding the non-hierarchical single process model and the non-hierarchical shared process model are the benchmarks to compare with the proposed hierarchical shared process Bayes model. However, within the entire methodology section (i.e., Sections 2 and 3), the definition of "single process model" is not clearly explained (only described with one sentence in line 331). Then later in the result section, the term "single process model" suddenly appeared which caused the reviewer to be confused. The reviewer went to the Appendix A to look for an answer but did success as the formulation of the "single hierarchical model" seems undescribed. The reviewer also finds it hard to understand why the link function is necessary to transform the parameter space of the standard deviation to the same as the sample space (Section 3.2). Is that a common practice for Bayesian models? The reviewer thinks a more detailed explanation will be helpful.*

AR:

We thank the reviewer for identifying this area for further clarification and have edited the manuscript accordingly. The manuscript has been comprehensively re-structured and re-written for clarity as recommended. In particular, the methodology section now includes a diagram showing the key steps of the framework proposed (see AR:1.2.6) and a data generation section is now included where the purpose of the simulated examples is discussed along with what comparisons are made. Additionally, further clarification on what the single process model refers to is provided in the results section and in the appendix section 'Shared and Single Process Model Comparison', see AR:1.3.3.

The purpose of the link function is also now clarified in the methodology: 'A link function is used for the case where the parameter space is not the same as the sample space for GP'. Further explained in the specific example in the appendix:

> In the case of the standard deviation the parameter space ($\sigma(s) \in \mathbb{R}_{>0}$) is not the same as the sample space of a GP ($\mathbb{R}$) and so a link function is applied $log(\sigma(s)) = \tilde{\sigma}(s) \in \mathbb{R}$. The transformed parameters are then modelled as being generated from GPs: $\tilde{\sigma}_Y(s) \sim \mathcal{GP}(m_{\tilde{\sigma}_Y}, k_{RBF}(s, s'|v_{\tilde{\sigma}_Y}, l_{\tilde{\sigma}_Y}))$ and $\tilde{\sigma}_B(s) \sim \mathcal{GP}(m_{\tilde{\sigma}_B}, k_{RBF}(s, s'|v_{\tilde{\sigma}_B}, l_{\tilde{\sigma}_B}))$.

Snippets from data generation section:

> Simulated examples are generated that highlight the advantage of two key features of the methodology over other approaches in the literature: modelling shared spatial covariance between the in situ data and climate model output through the inclusion of a shared generating latent process (Sect. 3.1) and the Bayesian hierarchical framework with uncertainty propagation (Sect. 3.2).

Snippet from results section:

> The shared latent process model presented in this paper is fit to the three non-hierarchical example scenarios, as discussed in Sect. 3.1. Input data for $\phi_Y(s_y)$ and $\phi_Z(s_z)$ are provided and the hyper-parameters for the latent GPs that generate the unbiased and biased components inferred. Comparisons in estimates of the hyper-parameters for the unbiased process ($m_{\phi_Y}$, $v_{\phi_Y}$ and $l_{\phi_Y}$) are made to the approach of only fitting to the in situ data, referred to here as the single process approach since the latent process generating the bias is not modelled. The difference between the shared and single process approaches is detailed further in appendix E.

**2.2. Minor Comments**

**2.2.1**

RC:

*1. The time series plot in Figure 11 exclusively illustrates the performance of the proposed Bayesian framework. The reviewer suggests including the performance of benchmark techniques in the same figure to facilitate comparisons.*

AR:

The performance of the proposed model for estimating the unbiased PDF parameters at the climate model locations $\phi_Y(s_z)$ is evaluated for non-hierarchical and hierarchical examples, with results presented in section 4.1 and 4.2 of the revised manuscript. Figures 6 and 7, as well as tables 3, 4 and 5 all illustrate performance. Figure 11 (now Fig. 8 in revised manuscript), illustrates how once $\phi_Y(s_z)$ is estimated, the uncertainty in the estimates can be propagated through quantile mapping to the final bias corrected time series. The manuscript has been comprehensively restructured to have a more conventional structure, with for example a single results section that hopefully further improves clarity to the reader about what each plot is showing. The methodology section is also updated to help provide further clarity on the key steps of the framework, see AR: 2.1.1.

**3. Referee 3**

**3.1. Main Comments**

**3.1.1**

RC:

*1. I definitely agree with other two reviewers that the manuscript is wordy and lengthy in its current shape. Specifically, there are some technical assumptions that are more or less mentioned in every single section (e.g. sum of GP is still a GP, bias is assumed to be independent from the in-situ observations, etc). I suggest authors revise their manuscript such that there can be a section that focuses on describing the assumptions they use in their method.*

AR:

It's agreed that the original manuscript had technical assumptions repeated in multiple sections. The manuscript has been revised such that a more conventional structure is adopted (see AR:2.1.1) where less repetition of assumptions is needed. In this new format, assumptions/modelling choices clearly described in the methodology, including time independence of the bias and independence between the unbiased and biased PDF parameter values. This is not in the form of a stand alone section as it is believed introducing each assumption at the corresponding relevant part of the methodology helps understanding.

> In a probabilistic framework, the in situ observations and climate model output are treated as realisations from latent spatiotemporal stochastic processes, denoted as $\{Y(s,t) : s \in \mathcal{S}, t \in \mathcal{T}\}$ and $\{Z(s,t) : s \in \mathcal{S}, t \in \mathcal{T}\}$ respectively. Stochastic processes are sequences of random variables indexed by a set, which in this case are the spatial and temporal coordinates in the domain $(\mathcal{S}, \mathcal{T})$. The observed data is then considered a realisation of the joint distribution over a finite set of random variables across the domain. For the purpose of evaluating the time-independent component of the climate model bias, the random variables are treated as independent and identically distributed across time. That is the collection of temporal data for a given spatial location can be considered as multiple realisations from the same random variable. The random variables for each location are distributed respectively as $Y(s) \sim f_Y(\phi_Y(s))$ and $Z(s) \sim f_Y(\phi_Z(s))$, where $\phi_Y(s)$ and $\phi_Z(s)$ represent the collection of parameters that describe the PDF. For example, if the PDF is approximated as normal then $\phi(s) = [\mu(s), \sigma(s)]$. The disparity between each of the PDF parameters for the in situ observations and climate model at each site then gives a measure of bias.

> The PDF parameters are each modelled as being generated from latent stochastic processes $\{\phi_Y(s)\}$ and $\{\phi_Z(s)\}$. The latent processes that generate the parameters for climate model are considered composed of two independent processes, one that also generates the equivalent parameters for the in situ observations and another that generates some bias, such that $\{\phi_Z(s)\} = \{\phi_Y(s)\} + \{\phi_B(s)\}$. The family of GPs are chosen for the latent processes.

**3.1.2**

RC:

*2. Given that the authors are about to clarify and shorten their assumptions, it is still of great interest whether the proposed method can work if the assumptions are slightly violated. Now the experiments are conducted using fully synthetic/simulated data that strictly follows the assumptions. I suggest authors prepare an example where it is not abundantly clear whether the assumptions still hold (a real data example), or at least generate synthetic data that intentionally violates the assumption. Some really strict assumptions in my opinion are:*

    *(i) the bias is independent of the in-situ observations*

*(ii) the bias is time independent*

*For example, if the authors can conduct still a synthetic experiment where the generated bias is slightly dependent on the time and climate model output, and run their algorithm against this case, this would make the proposed algorithm stronger. Another option for the authors is to refer to existing literature and argue that some of the assumptions hold for most cases in real applications.*

AR:

It's agreed that having some further discussion in the paper around the assumptions of the framework and remaining limitations regarding real-world scenarios is important. A paragraph discussing the assumptions and limitations of the methodology has been added to the discussion section of the paper, which hopefully addresses this adequately, see below and AR 1.1.6. The focus of the paper is primarily on providing a proof of concept around the approach of modelling underlying spatial covariance patterns and in fully propagating uncertainty to the end output. Addressing additional factors such as time independence of the bias, discussed extensively in papers such as Ehret et al., 2012; Maraun, 2012; Maurer et al., 2013; Chen et al., 2015, is outside the scope of the paper. Additionally, in cases where dependence is shown between the values of the unbiased PDF parameters of the in situ observations, while modelling this dependence would improve estimates, it is beyond the scope of this initial paper and a thorough analysis of this would be expected to over-complicate the results.

Section added to discussion on limitations:

> The simulated examples presented provide an initial proof of concept, although future studies validating the methodology against real-world applications are important for understanding the remaining limitations and areas for further development. The current primary limitation is expected to be that the underlying spatial covariance structures are assumed stationary. That is that the covariance length scale is assumed constant across the domain, whereas for real-world applications over large and complex topographic domains the length scale will be expected to change depending on the specific topography of the region. Further development of the methodology to incorporate non-stationary kernels would therefore be valuable, although is beyond the scope of this paper. Another important limitation to consider is the assumption that the bias is time independent. In situations where the bias varies gradually through time and uniformly across the domain, the methodology can be further developed such that the mean function of the GPs is modelled with a time dependency. If the bias varies in time non-uniformly across the domain, spatiotemporal GPs will need to be considered, which is again beyond the scope of this paper. Secondary limitations, include the assumption that the unbiased and biased components of the PDF parameter values are independent. In situations where there is a dependence between these components, the methodology presented is still expected to perform adequately, although information is lost by not modelling the dependency explicitly. Additionally, many real-world applications will necessitate specific model adjustments, such as incorporating a mean function dependent on factors like elevation and latitude. Finally, the computational complexity of the model is an important remaining consideration, with inference time of GPs scaling as the cube of the number of data points. Incorporating techniques from the literature such as using sparse variational GPs (SVGP) (Hensman et al., 2015), nearest-neighbor GPs (NNGP) (Datta et al., 2016) or upscaling the climate model output, while outside the scope of this paper, will aid computational performance under demanding real-world scenarios and will facilitate further model development.

**3.1.3**

RC:

*3. In terms of the structure of the manuscript, I personally have to understand the contribution of the manuscript until I start reading line 290 in page 12. It would help if the authors can make this clear very early in the manuscript.*

AR:

It is agreed that further clarity could be provided on the specific contributions of the methodology from an early point in the manuscript. A section has been added to the introduction to address this, as also mentioned in AR:2.1.2.

> The approach presented builds on that of Lima et al., 2021, which models the in situ observational data as generated from a GP and uses quantile mapping (Qian and Chang, 2021) to apply the correction to the climate model output. In Lima et al., 2021 the spatial covariance structure of the climate model output is not considered and uncertainty is not propagated to the final bias corrected time series. The novelty of the approach proposed here is that shared latent GPs are modelled between the climate model output and the in situ observational data, which aims to incorporate information from the physically realistic spatial patterns of the climate model output in predictions of the unbiased field. Additionally, uncertainty is propagated through the quantile mapping step, which results in uncertainty bands on the bias corrected output. The approach is developed with the focus of applying bias correction to regions with sparse in situ observations, such as over Antarctica, where capturing uncertainty in the correction is of key importance and where including data from all sources during inference is particularly valuable. Performance under simulated scenarios with differing data density and underlying covariance length scales is evaluated in this paper and the potential added value assessed when compared with the approach in Lima et al., 2021.

**3.2. Minor Comments**

**3.2.1**

**RC:**

*1. Is "1 process" just doing a GP for Y and "2 process" doing both GP for Y and Z? It is not very clear when I read through Figure 7 and its associated experiment section.*

AR:

It is agreed some further clarification on this could be provided. The '1 process' model does indeed just consider a single generating GP for the observational data $Y$. The '2 process' model considers generating processes for both $Y$ and the climate model output $Z$. As also mentioned in AR:1.3.3 a section has been added to the appendix, which the reader is referred to further clarify the difference between the approach suggested and that applied in Lima et al., 2021, see below.

**Appendix E: Shared and Single Process Model Comparison**

550 In the main paper comparisons are made made between the shared process and single process models. The shared process model is the hierarchical model proposed in this paper, while the single process model represents a similar approach taken from the literature, see Lima et al. (2021). The two models are shown in Fig. E1. In both models the random variables for the in-situ observations $Y(s)$ and climate model output $Z(s)$ have PDFs with the collection of parameters $\phi_Y(s)$ and $\phi_Z(s)$ respectively. In the case of the shared process model, $\phi_Z(s)$ is modelled as the sum of $\phi_Y(s)$ and some independent bias $\phi_B(s)$.

555 The parameters $\phi_Y(s)$ and the corresponding bias $\phi_B(s)$ are each themselves modelled over the domain as generated from Gaussian processes with hyper-parameters $\theta_Y$ and $\theta_B$. The unbiased parameters $\phi_Y(s)$ and hyper-parameters $\theta_Y$ are inferred from both the in situ data and climate model output. Posterior predictive estimates of $\phi_Y(s_z)$ are made by conditioning on both sets of data. In the case of the single process model, the PDF parameters for the climate model output and in situ observations are treated as independent. Only one latent GP is considered with the unbiased parameters $\phi_Y(s)$ and hyper-parameters $\theta_Y$

560 inferred from in situ observations alone. Posterior predictive estimates of $\phi_Y(s_z)$ are also made by conditioning on just in situ observations.

[Figure]

[Figure]

Shared Process Model                    Single Process Model

**Figure E1.** Plate diagram illustrating the difference between the shared process hierarchical model presented in this paper and the single process model that comparisons are made against.

**3.2.2**

**RC:**

*2. Line 115: "the their" is a typo*

AR:

Thanks, this sentence has been removed as part of the re-writing and re-structuring.

**3.2.3**

**RC:**

*3. I personally find that Appendix A is easier to follow than some of the text in the main body. Authors may want to consider*

*restructuring the manuscript*

AR:

Thanks for this feedback, it is agreed that using parts of section A in the main body will further help with clarity. We've re-written the methodology to include elements of this section (A1-A11) as suggested. The posterior and posterior predictive formulations (A12-A29) are kept in the appendix to keep the main part of the paper more approachable for researchers without an expertise in statistics.

**References**

Carter, Jeremy, Amber Leeson, Andrew Orr, Christoph Kittel, and J. Melchior van Wessem (Sept. 23, 2022). "Variability in Antarctic surface climatology across regional climate models and reanalysis datasets". In: *The Cryosphere* 16.9. Publisher: Copernicus GmbH, pp. 3815–3841. ISSN: 1994-0416. DOI: 10.5194/tc-16-3815-2022. URL: https://tc.copernicus.org/articles/16/3815/2022/ (visited on 10/04/2022).

Chen, Jie, François P. Brissette, and Philippe Lucas-Picher (2015). "Assessing the limits of bias-correcting climate model outputs for climate change impact studies". In: *Journal of Geophysical Research: Atmospheres* 120.3. _eprint: https://onlinelibrary.wiley.com/doi/pdf/ pp. 1123–1136. ISSN: 2169-8996. DOI: 10.1002/2014JD022635. URL: https://onlinelibrary.wiley.com/doi/abs/10.1002/2014JD022635 (visited on 03/21/2024).

Datta, Abhirup, Sudipto Banerjee, Andrew O. Finley, and Alan E. Gelfand (Apr. 2, 2016). "Hierarchical Nearest-Neighbor Gaussian Process Models for Large Geostatistical Datasets". In: *Journal of the American Statistical Association* 111.514. Publisher: Taylor & Francis _eprint: https://doi.org/10.1080/01621459.2015.1044091, pp. 800–812. ISSN: 0162-1459. DOI: 10.1080/01621459.2015.1044091. URL: https://doi.org/10.1080/01621459.2015.1044091 (visited on 03/24/2024).

Ehret, U., E. Zehe, V. Wulfmeyer, K. Warrach-Sagi, and J. Liebert (Sept. 21, 2012). "HESS Opinions "Should we apply bias correction to global and regional climate model data?"" In: *Hydrology and Earth System Sciences* 16.9. Publisher: Copernicus GmbH, pp. 3391–3404. ISSN: 1027-5606. DOI: 10.5194/hess-16-3391-2012. URL: https://hess.copernicus.org/articles/16/3391/2012/ (visited on 07/29/2021).

Hensman, James, Alexander G Matthews, Maurizio Filippone, and Zoubin Ghahramani (2015). "MCMC for Variationally Sparse Gaussian Processes". In: *Advances in Neural Information Processing Systems*. Vol. 28. Curran Associates, Inc. URL: https://proceedings.neurips.cc/paper_files/paper/2015/hash/6b180037abbebea991d8b1232f8a8ca9-Abstract.html (visited on 03/24/2024).

Hoffman, Matthew D. and Andrew Gelman (2014). "The No-U-Turn Sampler: Adaptively Setting Path Lengths in Hamiltonian Monte Carlo". In: *Journal of Machine Learning Research* 15.47, pp. 1593–1623. ISSN: 1533-7928. URL: http://jmlr.org/papers/v15/hoffman14a.html (visited on 10/11/2023).

Lima, Carlos H. R., Hyun-Han Kwon, and Yong-Tak Kim (June 1, 2021). "A Bayesian Kriging model applied for spatial downscaling of daily rainfall from GCMs". In: *Journal of Hydrology* 597, p. 126095. ISSN: 0022-1694. DOI: 10.1016/j.jhydrol.2021.126095. URL: https://www.sciencedirect.com/science/article/pii/S0022169421001426 (visited on 08/16/2021).

Maraun, D. (2012). "Nonstationarities of regional climate model biases in European seasonal mean temperature and precipitation sums". In: *Geophysical Research Letters* 39.6. _eprint: https://onlinelibrary.wiley.com/doi/pdf/10.1029/2012GL051210. ISSN: 1944-8007. DOI: 10.1029/2012GL051210. URL: https://onlinelibrary.wiley.com/doi/abs/10.1029/2012GL051210 (visited on 11/16/2021).

Maurer, E. P., T. Das, and D. R. Cayan (June 7, 2013). "Errors in climate model daily precipitation and temperature output: time invariance and implications for bias correction". In: *Hydrology and Earth System Sciences* 17.6. Publisher: Copernicus GmbH, pp. 2147–2159. ISSN: 1027-5606. DOI: 10.5194/hess-17-2147-2013. URL: https://hess.copernicus.org/articles/17/2147/2013/ (visited on 03/21/2024).

Phan, Du, Neeraj Pradhan, and Martin Jankowiak (Dec. 24, 2019). *Composable Effects for Flexible and Accelerated Probabilistic Programming in NumPyro*. DOI: `10.48550/arXiv.1912.11554`. URL: `http://arxiv.org/abs/1912.11554` (visited on 10/18/2023).

Qian, Weijia and Howard H. Chang (Feb. 2021). "Projecting Health Impacts of Future Temperature: A Comparison of Quantile-Mapping Bias-Correction Methods". In: *International Journal of Environmental Research and Public Health* 18.4, p. 1992. ISSN: 1661-7827. DOI: `10.3390/ijerph18041992`. URL: `https://www.ncbi.nlm.nih.gov/pmc/articles/PMC7922393/` (visited on 07/04/2023).

---

## Author Response (AR2)

**Authors' Response to Reviews of**

**Bias Correction of Climate Models using a Bayesian Hierarchical Model**

J. Carter
*Geoscientific Model Development,*
* * *
**RC:** *Reviewers' Comment*,    AR: Authors' Response,    ☐ Manuscript Text

Thank you for the further review of the revised manuscript. We are very grateful for your time and have worked hard to incorporate the additional feedback into the revised manuscript. The response is detailed here and we hope that you find the further enhancements to the paper satisfactory. A revised manuscript is provided, along with a LaTex-diff document highlighting changes.

**1.  Referee 2 Further Comments**

**RC:**
*1. I appreciate that the proposed method would potentially help correct the bias of climate model outputs by preserving the covariance structure. Extensive experimentation on synthetic data under different distributional assumptions has also helped me better understand how the framework improves over a single-process model. However, it has not been convincingly argued that the synthetic experiments represent real-world scenarios or that the synthetic data experiments reveal how the model may perform on real climate model data. It is still unclear if the proposed methodology would be useful in real-world scenarios.*

AR:
We thank the reviewer for their constructive comment and have edited the manuscript accordingly to further highlight the applicability of the methodology to real-world data sets. To start, additional writing (highlighted in blue below) is included in Sect. 3.1 of the manuscript further describing how the simulated scenarios are related to different potential real-world cases, using the study of Lima et al., 2021 as an example.

Relevant text from data generation section (added text highlighted in blue):

> Data is generated for three scenarios chosen to represent different potential real-world situations, illustrated in Fig. 3. The first scenario (Fig. 3a) represents an example case where it is expected that there is ample data provided in the form of in situ observations to capture the features of the underlying complete realisation of $\phi_Y$ without significant added value provided from inclusion of the climate model output during inference. The second scenario (Fig. 3b) is an adjustment where the in situ observations are relatively sparse and the underlying bias is relatively smooth. In this situation the climate model output should provide significant added value in estimating $\phi_Y$ across the domain since it is only afflicted by a comparatively simple bias that is easy to estimate. The final scenario (Fig. 3c) also involves sparse in situ observational data but with a reduced smoothness of the bias compared to the other scenarios. In this scenario the climate model output should provide added value in estimating $\phi_Y$ across the domain but this will be limited compared to scenario two due to the difficulty of disaggregating the components and estimating the comparatively more complex bias.
>
> In practice, real-world datasets are likely to be a combination of these scenarios. For example, the methodology in Lima et al., 2021 is applied to bias correcting precipitation over a domain covering South Korea and the surrounding ocean. Over the land, there is a sufficient spatial density of observational rainfall gauges to adequately capture the spatial features

> of the unbiased underlying field from the observations alone (similar to scenario A). Over the ocean, rainfall gauges are very sparse and so its important to consider the spatial patterns observed from the climate model output (similar to scenario B). Not accounting for the spatial features seen in the climate model output over the ocean results in undesirable extrapolation over this region, as seen in the results presented in Lima et al., 2021. This undesirable property is something that is addressed by the methodology proposed in this paper, as illustrated by results for scenario B given in Sect. 4.1.

In addition to this, further writing has been included in the introduction detailing the flexibility and applicability of using GPs for modelling real-world datasets. Reference is made to papers from relevant text in the literature, including Zhang et al., 2021, Lima et al., 2021 and Wang and Chaib-draa, 2017.

Relevant added text to introduction:

> While simple simulated scenarios are focused on in this paper, the applicability of GPs for modelling complex spatial patterns seen in real-world climatology is already illustrated in Zhang et al., 2021 and Lima et al., 2021. The non-parametric nature of GPs makes the model flexible and able to capture complex non-linear spatial relationships. Additionally, features of GPs such as uncertainty estimation, sensible extrapolation, kernel customisation and the ability to produce accurate predictions with limited data are desirable for real-world case studies. Finally, advancements in approximate inference methods have improved the scalability of GPs, improving the applicability to large climate data sets, as demonstrated in Wang and Chaib-draa, 2017. In addition to the main results presented in Sect. 4, to further demonstrate the flexibility and applicability of the methodology presented in this paper to potential real-world scenarios, some additional simulated scenarios are created with added complexity and results presented in appendix G. These additional scenarios test the robustness of the model to potential real-world situations where not all the assumptions of the model will necessarily completely hold.

Finally, a whole new section of the appendix is created to illustrate the applicability of the methodology under additional scenarios with added complexity and where some of the assumptions of the model are partially broken, as might be the case in real-world scenarios. The results from this section show the model proposed is robust and behaves desirably with potentially challenging features of real-world datasets. Despite this, importance is placed on how 
[revised manuscript text omitted]
". In: *Journal of Hydrology* 597, p. 126095. ISSN: 0022-1694. DOI: 10.1016/j.jhydrol.2021.126095. URL: https://www.sciencedirect.com/science/article/pii/S0022169421001426 (visited on 08/16/2021).

Wang, Yali and Brahim Chaib-draa (Jan. 1, 2017). "An online Bayesian filtering framework for Gaussian process regression: Application to global surface temperature analysis". In: *Expert Systems with Applications* 67, pp. 285–295. ISSN: 0957-4174. DOI: 10.1016/j.eswa.2016.09.018. URL: https://www.sciencedirect.com/science/article/pii/S095741741630495X (visited on 05/14/2024).

Zhang, Yongshun, Miao Feng, Weimin Zhang, Huizan Wang, and Pinqiang Wang (July 1, 2021). "A Gaussian process regression-based sea surface temperature interpolation algorithm". In: *Journal of Oceanology and Limnology* 39.4, pp. 1211–1221. ISSN: 2523-3521. DOI: 10.1007/s00343-020-0062-1. URL: https://doi.org/10.1007/s00343-020-0062-1 (visited on 05/14/2024).